# ASTRO: Teaching Language Models to Reason by Reflecting and Backtracking In-Context

## Abstract

We introduce ASTRO, the "Autoregressive Search-Taught Reasoner", a framework for training language models to reason like search algorithms, explicitly leveraging self-reflection, backtracking, and exploration in their outputs. Recently, training large language models (LLMs) via reinforcement learning (RL) has led to the advent of reasoning models with greatly enhanced reasoning capabilities. Open-source replications of reasoning models, while successful, build upon models that already exhibit strong reasoning capabilities along with search behavior observed even before RL. As a result, it is yet unclear how to boost the reasoning capabilities of other models including Llama 3. ASTRO teaches such models to internalize structured search behavior through a synthetic dataset derived from Monte Carlo Tree Search (MCTS) over mathematical problem-solving trajectories. By converting search traces into natural language chain-of-thoughts that capture both successes and recoveries from failure, ASTRO bootstraps models with a rich prior for exploration during RL. We finetune our model on these search-derived traces and further apply RL. We apply ASTRO to the Llama 3 family of models and achieve absolute performance gains of 16.0% on MATH-500, 26.9% on AMC 2023, and 20.0% on AIME 2024, especially improving upon challenging problems that require iterative correction. Our results demonstrate that search-inspired training offers a principled way to instill robust reasoning capabilities into open LLMs.

## 1 Introduction

Training large language models (LLMs) via reinforcement learning (RL) has greatly improved their reasoning capabilities, leading to the advent of reasoning models such as OpenAI o1 (OpenAI, 2024), DeepSeek-R1 (DeepSeek-AI, 2025) or Gemini 2.5 (Google, 2025). A prominent feature of reasoning models is their ability to iteratively refine their outputs with a behavior similar to *search* – a process which involves reflecting on their own outputs and backtracking to a previous state (Xiang et al., 2025). While open-source replications of reasoning models achieve notable performance improvements, they rely on distillation from existing reasoning models (Li et al., 2025; Muennighoff et al., 2025) or direct RL (Hu et al., 2025; Yu et al., 2025) from LLMs that (1) already contain reflective behavior and strong reasoning capabilities (Chang et al., 2025; Liu et al., 2025), and (2) exhibit spurious performance gains from incorrect or noisy reward signals during RL (Lv et al., 2025; Shao et al., 2025). Hence it is unclear from a scientific perspective how reasoning models can be built from other LLMs that do not exhibit the aforementioned behavior, such as Llama 3 (AI at Meta, 2024).

We introduce **ASTRO**, the "Autoregressive Search-Taught Reasoner", a framework that systematically infuses search-like behavior into language models *ab initio* to improve their reasoning capabilities. The fundamental principle guiding ASTRO is *search*, where our policy explores the solution space by selecting actions, reflecting on its own solution, and backtracking to a previous step if needed. ASTRO trains language models to perform **autoregressive search** – instead of using external search scaffolds such as beam search to solve reasoning problems, ASTRO internalizes the search procedure and generates entire search trajectories, including reflections and backtracks, in a single inference pass. Models trained using ASTRO exhibit improved reasoning abilities by frequently re-evaluating their solutions and backtracking until they reach a final answer of high confidence. Such models also generate structured reasoning traces that can be mapped to a directed graph with each vertex representing a discrete reasoning step, allowing for a richer understanding of their reasoning processes.

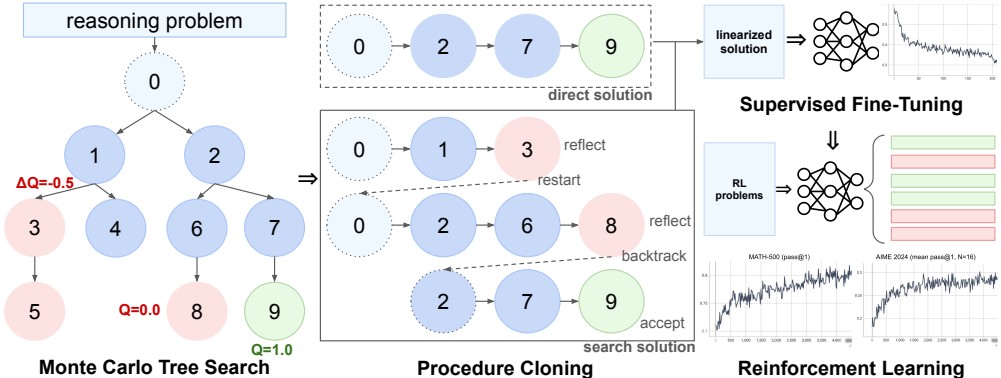

Figure 1: An overview of ASTRO. Given a math reasoning problem, we first perform Monte Carlo Tree Search (MCTS) and obtain a search tree where each node contains a discrete reasoning step with its associated Q-value. We then linearize the visited sequence of nodes into a solution that integrates backtracking and self-reflection in natural language. Then, we perform supervised fine-tuning (SFT) on the search-integrated solutions and bootstrap our policy to perform autoregressive search. Finally, we further improve the policy's search and reasoning capabilities with reinforcement learning (RL).

ASTRO operates in three stages: (1) search trajectory generation, (2) supervised fine-tuning and (3) reinforcement learning. We bootstrap our models with search behavior by generating search traces to be used for training data via **procedure cloning** (Yang et al., 2022; Laskin et al., 2022) – we perform search with our LM policy to explore over different solution trajectories for each math problem, and we train our policy *without* using search scaffolds at test time to predict the entire sequence of actions that end with a successful terminal state. ASTRO provides beneficial priors for RL by systematically injecting self-reflection and backtracking patterns to the search traces via procedure cloning.

First, we generate synthetic data, also called the *cold-start data* (DeepSeek-AI, 2025; Qwen, 2025), to instill in-context search priors to our models. We use Monte Carlo Tree Search (MCTS) to explore the solution space of challenging math problems and build search trees with diverse reasoning traces. Then, we linearize each search tree into a sequence of visited nodes, which we convert into a natural language chain-of-thought (CoT, Wei et al. (2022)) that integrates self-reflection and backtracking. Then, we sample about 36K high-quality CoT solutions across three open-source math datasets.

We then perform supervised fine-tuning (SFT) to infuse autoregressive search behavior into the Llama 3 family of models (AI at Meta, 2024). After fine-tuning for just one epoch, our SFT checkpoint based on `llama-3.1-70b-instruct` achieves 69.6% on MATH-500, 55.0% on AMC 2023 and 13.3% on AIME 2024, and outperforms its counterpart trained on the same set of problems without search priors. Our qualitative analyses show that even simply performing SFT with high-quality search traces can infuse search capabilities into a language model.

Finally, we perform reinforcement learning (RL) on our models to further improve their reasoning capabilities. Our training prompts are derived from open-source math problems of moderate to high difficulties. We use a modified form of Group Relative Policy Optimization (GRPO, Shao et al. (2024)) to update our policies. After RL, our policy based on `llama-3.1-70b-instruct` achieves 81.8% in MATH-500, 64.4% in AMC 2023 and 30.0% in AIME 2024 (pass@1). We show that our model trained end-to-end using ASTRO outperforms its counterpart similarly optimized with RL but initialized from a SFT checkpoint trained without search priors – this demonstrates the importance of leveraging self-reflection and backtracking as priors for improving reasoning via RL.

## 2 SEARCH TRAJECTORY GENERATION

ASTRO begins by generating a dataset of search traces, expressed as long chain-of-thoughts (Wei et al., 2022) that encode self-reflection and backtracking in natural language, via procedure cloning. To this end, we first obtain search trees for each math problem using Monte Carlo Tree Search (MCTS) in a stepwise manner, strategically balancing exploration and exploitation with verifier-based rewards to obtain high-quality solutions exploring diverse reasoning traces (Section 2.2).

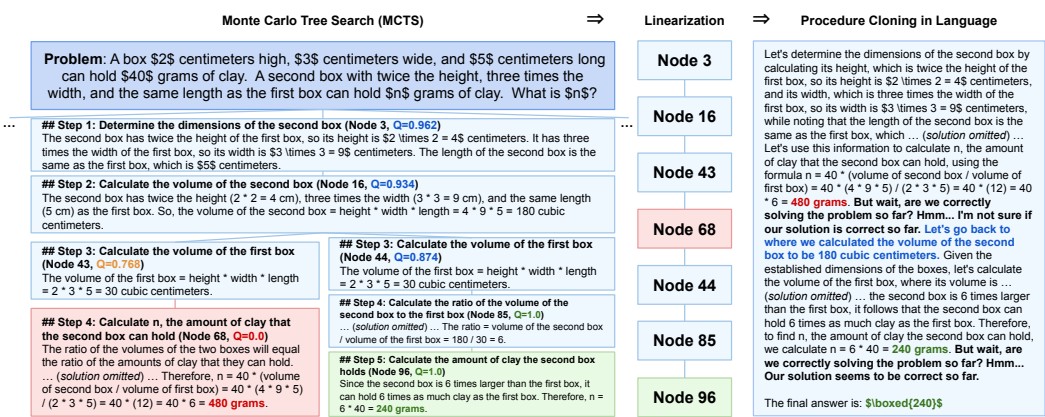

Figure 2: Example of search trajectory generation via procedure cloning. We perform search through the solution space via MCTS and track terminal nodes with both incorrect and correct answers. Then, we linearize the search tree and integrate backtracking from an incorrect terminal node to its common ancestor with the correct terminal node. We rewrite this node sequence into a long chain-of-thought.

We then linearize the search trees into sequences of nodes that explore intermediate nodes with incorrect answers until arriving at the correct answer (Section 2.3). Finally, we translate each node sequence into a long CoT that integrates self-reflection and backtracking (Section 2.4). Our dataset encodes beneficial in-context search priors for training LLMs to solve challenging math problems via SFT and RL (Section 3). Refer to Figure 2 for a visual example of our long-CoT generation pipeline.

## 2.1 PROBLEM FORMULATION AND OVERVIEW

**Problem formulation.** Our data generation setup is a Markov Decision Process (MDP) (Puterman, 1994), where the LM functions as the policy $\Pi_{LM}$ and explores the solution space to the input $\mathbf{x}$, while obtaining rewards from a verifier $\mathcal{V}$. Here we assume that $\Pi_{LM}$ solves math problems in a stepwise manner, where each step $s_t$ represents a sequence of tokens $y_1 \cdots y_{|s_t|}$ encapsulating a minimal unit of reasoning that solves $\mathbf{x}$. Then, each state $S_t$ is a combination of the input prompt and the generated steps, i.e. $S_t = (\mathbf{x}, s_0, \cdots, s_t)$. Meanwhile, the action $a_{t+1}$ is the next step $s_{t+1}$ taken by $\Pi_{LM}$.

Given this setup, we teach $\Pi_{LM}$ to predict a sequence of states $(S_0 \cdots S_{end})$ to perform in-context search and explore the solution space until it terminates its search at $S_{end}$ with the correct final answer.

**Overview.** We generate training data for ASTRO in three main stages outlined below:

1. For each $\mathbf{x}$ we generate a search tree $T$, where each node $n_i$ represents the state $S_i$ and each edge $(n_i, n_j)$ represents the action $a_j$, i.e. the next step $s_j$ taken from $S_i$ to $S_j$, using MCTS to explore the solution space based on verifier rewards from rollouts (Section 2.2).

2. We linearize $T$ into a sequence of nodes $L = (n_0, \cdots, n_{end})$, a subsequence of the history of nodes visited by $\Pi_{LM}$ until arriving at $n_{end}$, the terminal node with the correct answer. Some adjacent pairs of nodes $(n_t, n_{t+1})$ in $L$ are such that $n_{t+1}$ is an ancestor of $n_t$ in $T$, which corresponds to self-reflection and backtracking during the search procedure (Section 2.3).

3. We translate $L$ into a chain-of-thought solution $\mathbf{y} = (y_0, \cdots, y_{end})$ integrating self-reflection and backtracking in natural language, and we add $(\mathbf{x}, \mathbf{y})$ to our final dataset (Section 2.4).

## 2.2 MONTE CARLO TREE SEARCH

We use our language model policy $\Pi_{LM}$ to obtain a search tree with diverse solution traces to each input $\mathbf{x}$ by running Monte Carlo Tree Search (MCTS). By using MCTS, we explore a diverse solution space while balancing exploration and exploitation with reliable guidance from reward signals obtained from full rollouts. Here, we prompt $\mathbf{x}$ to elicit stepwise solutions from $\Pi_{LM}$, and assign reward scores with our verifier $\mathcal{V}$ to compare the predicted answer with the correct answer. We provide detailed equations for MCTS in Section A.1 of the Appendix (Equations 2, 3, 4 and 5).

**Selection.** At state $S_t$ with $k$ actions generated by $\Pi_{\text{LM}}$, we select the most promising node from which to further perform tree search. We use the Predictor+Upper Confidence bounds applied to Trees (PUCT, Silver et al. (2016)) to balance exploration and exploitation, as shown in Equation 2.

**Expansion.** From state $S_t$, $\Pi_{\text{LM}}$ takes $\mathbf{x}$ and the sequence of steps $(s_0, \cdots, s_t)$ as the input, and first samples $k$ next actions for solving $\mathbf{x}$. For each action, we sample $M$ rollouts and score them using $\mathcal{V}$. Then, we average the rollout scores for each new action $a_i$ ($i \in [1...k]$) to compute the reward scores for the new states. We add a new node $n_{t+1}$, associated with $S_{t+1}$, to $T$ (Eq. 3).

**Backpropagation.** We backpropagate the reward scores from the leaf node to the root node to recursively update their Q-values. The updates consist of (1) incrementing the visit count of each state (Eq. 4), and (2) updating the Q-values of each (state, action) pair using the Q-values and visit counts of the children nodes of $S_{t+1} = (S_t, a)$, along with the reward score $R(S_{t+1})$ (Eq. 5).

We repeat these steps for 32 iterations, using `llama-3.3-70b-instruct` as $\Pi_{\text{LM}}$ with $k = 8$ actions and $M = 16$ rollouts, $c_{\text{puct}} = 1.0$ and a maximum tree depth of 50.

## 2.3 SEARCH TREE LINEARIZATION

We convert the search tree $T$ into a sequence of states $(S_0, \cdots, S_{\text{end}})$ exploring correct and incorrect reasoning steps until arriving at the correct final answer, by sampling $L = (n_0, \cdots, n_{\text{end}})$ from $T$.

In this process, we maintain the invariants that (1) *the sequence ends with a correct terminal node*, and (2) *the sequence may contain incorrect terminal nodes, but no two nodes in L may contain the same incorrect answer*. With these invariants, we ensure that our model learns not to repeat the same incorrect answer, and learns to rectify its incorrect reasoning with a high-quality, correct solution.

Our linearization procedure is detailed in Algorithm 1. First, we perform depth-first search on $T$ to identify terminal nodes with correct and incorrect answers, and we collect a subset of correct terminal nodes with high-quality reasoning steps based on the policy's self-evaluation (Appendix A.2). We randomly sample a correct terminal node, and a subset of $k$ terminal nodes that contain distinct incorrect answers. Then, we iterate through these terminal nodes, gather the sequence from the root node to each terminal node, and sequentially merge the sequences. We thereby obtain a linearized sequence $L = (n_0, \cdots, n_{\text{end}})$ with $k$ backtracking instances, where $n_{\text{end}}$ contains the correct answer.

---

**Algorithm 1** Search Tree Linearization.

---

**Input:** input $\mathbf{x}$, answer $a$, number of backtracks $k$, policy $\Pi$, search tree $T$ with root node $n_0$
$\Psi, \Lambda$: set of terminal nodes with correct and incorrect answers in $T$
$\Psi'$: subset of $\Psi$ that contains high-quality solutions
$\Psi, \Lambda \leftarrow \text{DFS(root)}, \Psi' \leftarrow \{\}, L \leftarrow [\,]$
\# Collect all terminal nodes with high-quality, correct solutions
**for** $n_\psi \in \Psi$ **do**
$\quad S \leftarrow \text{concat\_steps\_from\_root}(n_\psi)$
$\quad$ **if** $\Pi(\text{self\_eval\_prompt}, S) == 1$ **then** $\text{add}(\Psi', n_\psi)$
**end for**
\# Sample a correct node, and a subset of $k$ nodes with distinct incorrect answers
$n_{\psi*} \leftarrow \text{sample}(\Psi'), \Lambda_k \leftarrow \text{sample\_with\_unique\_answers}(\Lambda, k)$
\# Build the linearized sequence $L$ by tracking the $k$ incorrect nodes and $n_{\psi*}$
**for** $n \in \Lambda_k \cup \{n_{\psi*}\}$ **do**
$\quad l \leftarrow \text{gather\_nodes\_from\_root}(n)$
$\quad L \leftarrow \text{merge\_sequences}(L, l - L)$
**end for**
**Output:** the linearized sequence of nodes $L$

---

## 2.4 PROCEDURE CLONING IN LANGUAGE

We convert $L = (n_0, \cdots, n_{\text{end}})$ into a chain-of-thought solution $\mathbf{y}$ by processing the nodes sequentially. Given the node $n_t$ which contains a reasoning step $s_t$ and the existing solution $\mathbf{y}_{1:t-1}$ at timestep $t$, we few-shot prompt $\Pi_{\text{LM}}$ to generate $y_t$. We address two cases (Appendix A.3):

*Case 1: $n_t$ is a child of $n_{t-1}$.* We simply convert the step $s_t$ into $y_t$ which continues from $\mathbf{y}_{1:t-1}$.

*Case 2: $n_t$ is an ancestor of $n_{t-1}$.* In this case, we first add a hard-coded self-reflection phrase `But wait, are we solving the problem correctly so far? Hmm...`, followed by `I'm not sure if we're solving the problem correctly so far.` Then, we few-shot prompt $\Pi_{\text{LM}}$ to generate a sentence that describes backtracking (if $n_t$ is a non-root node – `Let's go back to where we ...`) or restart (if $n_t$ is a root node – `Let's restart our solution from the beginning ...`).

In this way, our solution integrates self-reflection followed by a continuation of the solution or backtracking to a previous step. We repeat this procedure for all of our linearized node sequences.

## 3 LEARNING TO SEARCH

We use the long-CoT search traces from Section 2 to train our models to perform in-context search. We first use supervised fine-tuning (SFT) to infuse search priors into our models (Section 3.1), and then reinforcement learning (RL) to further improve their reasoning capabilities (Section 3.2).

### 3.1 SUPERVISED FINE-TUNING

We perform supervised fine-tuning (SFT) on a subset of the search trajectories collected in Section 2 to instill self-reflection and backtracking priors into the Llama 3 family of models (AI at Meta, 2024). In this stage, we focus on curating a relatively small amount of *high-quality* search trajectories in order to carefully infuse our policy with helpful priors to be exploited later during RL.

**Dataset.** We use three open math datasets: `MATH-train` (Hendrycks et al., 2021), and the `AMC/AIME` and `AoPS-forum` subsets of `NuminaMath` (Li et al., 2024).

We build a total of 20.7K search trees and identify 14.0K search trees with at least one high-quality, correct solution. After linearization, we curate a total of 105K long-CoT solutions. For each math problem we sample one solution without any backtracking ($k = 0$), and three solutions with at least one self-reflection and backtracking ($k \geq 1$) and obtain a total of 36.1K solutions. Refer to Tables 3 and 4 in Appendix B for more information regarding the composition of our main SFT dataset.

**Training.** We perform SFT on `llama-3.1-70b-instruct`, and we simply train on all tokens from each example without any masking. Refer to Appendix B for more details on our SFT procedure.

### 3.2 REINFORCEMENT LEARNING

Based on our fine-tuned models that perform autoregressive search, we further leverage the models' search priors to improve their reasoning capabilities through reinforcement learning (RL).

**Dataset.** We use open-source math datasets for our training prompts: `AIME 1983-2023` which consist of AIME problems from the years 1983 to 2023, `MATH-4500` which is separated from the `MATH-500` evaluation set, along with `MATH-train` and the `AMC/AIME` and `AoPS-forum` subsets of `NuminaMath`.

To curate the final dataset, we use the model from Section 3.1 to generate $N = 64$ solutions for each problem, and we compute the model's *pass rate*, i.e. the fraction of correct solutions generated. We curate problems where the pass rate ranges from 1% to 75% to exclude problems that are too easy or difficult for the policy, and focus on problems of moderate to high difficulty. Refer to Table 5 in Appendix C for details of various statistics for our RL datasets.

**Training.** We train our models using Group Relative Policy Optimization (GRPO, Shao et al. (2024)) while computing advantages based on subtracting the prompt-level mean reward and removing the standard deviation (Liu et al., 2025), and setting the KL penalty to zero. Given our RL dataset $\mathcal{D}$ with training examples $(\mathbf{x}, \mathbf{y})$ where $\mathbf{x}$ is the input prompt and $\mathbf{y}$ is the correct answer, a batch of solutions $\mathbf{S}$ generated for each $\mathbf{x}$, the verifier $\mathcal{V}$ which assigns a binary correctness score to each solution $\mathbf{s} \in \mathbf{S}$ based on $\mathbf{y}$, as well as our policy $\pi_\theta$, we optimize its parameters $\theta$ such that the following holds:

$$\max_\theta \mathbb{E}_{(\mathbf{x},\mathbf{y}) \sim \mathcal{D}} \left[ \mathbb{E}_{\mathbf{s} \in \mathbf{S} \sim \pi_\theta(\cdot|\mathbf{x})} \left[ \mathcal{V}(\mathbf{s}, \mathbf{y}) - \frac{1}{\|\mathbf{S}\|} \sum_{\mathbf{s}' \in \mathbf{S}} \mathcal{V}(\mathbf{s}', \mathbf{y}) \right] \right] \tag{1}$$

| Checkpoint | MATH-500 | AMC 2023 | | AIME 2024 | |
|---|---|---|---|---|---|
| | pass@1 | pass@1 | maj@8 | pass@1 | maj@8 |
| Llama-3.1-70B-Instruct | 65.8 | 37.5 | 47.5 | 10.0 | 16.7 |
| Llama-3.3-70B-Instruct | 75.8 | 57.5 | 60.0 | 26.7 | 30.0 |
| Llama-3.1-70B-Instruct (SPOC) | 77.4 | 52.5 | - | 23.3 | - |
| Llama-3.1-70B-Instruct (Step-KTO) | 76.2 | 60.0 | 67.5 | 16.7 | 20.0 |
| **Llama-3.1-70B-ASTRO-SFT** | 69.6 | 51.9 | 63.0 | 16.3 | 24.7 |
| **Llama-3.1-70B-ASTRO-RL** | **81.8** | **64.4** | **68.8** | **30.0** | **32.0** |
| Llama-3.3-70B-Instruct (SPOC) | 77.8 | 70.0 | - | 23.3 | - |
| Llama-3.3-70B-Instruct (Step-KTO) | 79.6 | 70.0 | 75.0 | 30.0 | 33.3 |

Table 1: Main results. **Llama-3.1-70B-ASTRO-SFT** outperforms `llama-3.1-70b-instruct`. **Llama-3.1-70B-ASTRO-RL** outperforms `llama-3.3-70b-instruct`, as well as SPOC and Step-KTO based on `llama-3.1-70b-instruct`, and sometimes outperforms or matches SPOC and Step-KTO based on `llama-3.3-70b-instruct`. Note that our pass@1 scores for AMC 2023 and AIME 2024 evaluations are averaged over 16 different runs, while the baseline scores are not.

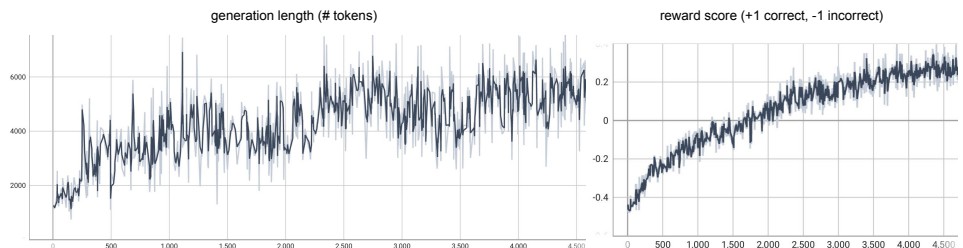

Figure 3: Training curves detailing how **Llama-3.1-70B-ASTRO-RL** behaves during RL training. **(Left)** The length of the model's CoT during training. The generation length gradually increases from 1,600 tokens to almost 6,000 tokens as training progresses. **(Right)** The reward score assigned to the policy during RL, with correct and incorrect answers being assigned scores of +1 and -1, respectively.

## 4 EXPERIMENTS AND RESULTS

### 4.1 EXPERIMENTAL SETUP

**Training.** We train our models with datasets in Sections 3.1 and 3.2 for SFT and RL. For our SFT experiments, we fine-tune `llama-3.1-70b-instruct` on our synthetic dataset with 36.1K search trajectories for 1 epoch only, in order to prevent overfitting to the SFT dataset. For our RL experiments, we further optimize the checkpoints we obtain from the SFT stage using the objective presented in Equation 1. Refer to Appendix B.2 and C.2 for more details on our training hyperparameters.

**Evaluation.** We evaluate our models on three widely-used math evaluation benchmarks – MATH-500 (Hendrycks et al., 2021), AMC (American Mathematics Competition) 2023 and AIME (American Invitational Mathematics Examination) 2024. Due to the lower number of examples present in the AMC 2023 and AIME 2024 benchmarks, we compute the pass@1 score by averaging across $N = 16$ generations. We also compute the maj@8 score by sampling a random subset of eight outputs over ten simulations and averaging the scores obtained for the majority answers across the ten simulations.

**Baselines.** Our baselines are `llama-3.1-70b-instruct`, which employs SFT and Direct Preference Optimization (DPO, Rafailov et al. (2023)) for post-training and `llama-3.3-70b-instruct`, which is further trained via online RL. Moreover, we compare with two methods based on RL or self-correction that improve Llama 3's mathematical reasoning: Step-KTO(Lin et al., 2025) which augments KTO (Ethayarajh et al., 2024) with PRMs, and SPOC (Zhao et al., 2025) which learns to self-correct and verify in-context with traces obtained via Pair-SFT (Welleck et al., 2023) and RL.

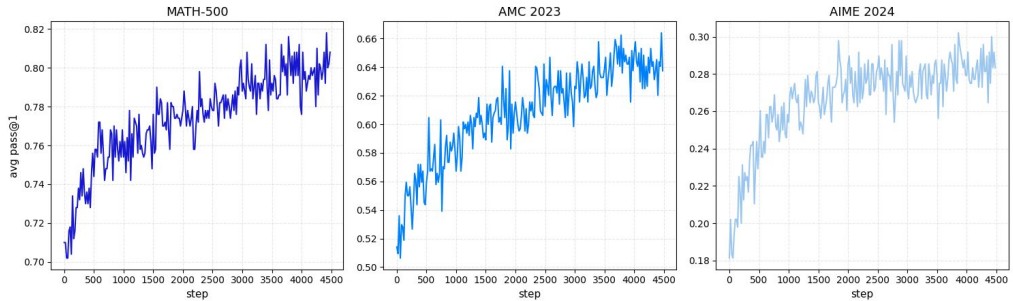

Figure 4: Performance on MATH-500, AMC 2023 and AIME 2024 while training **Llama-3.1-70B-Astro-RL** via RL. The evaluation metrics improve steadily across all three benchmarks. We use the pass@1 metric, with the AMC 2023 and AIME 2024 metrics being averaged over 16 different runs.

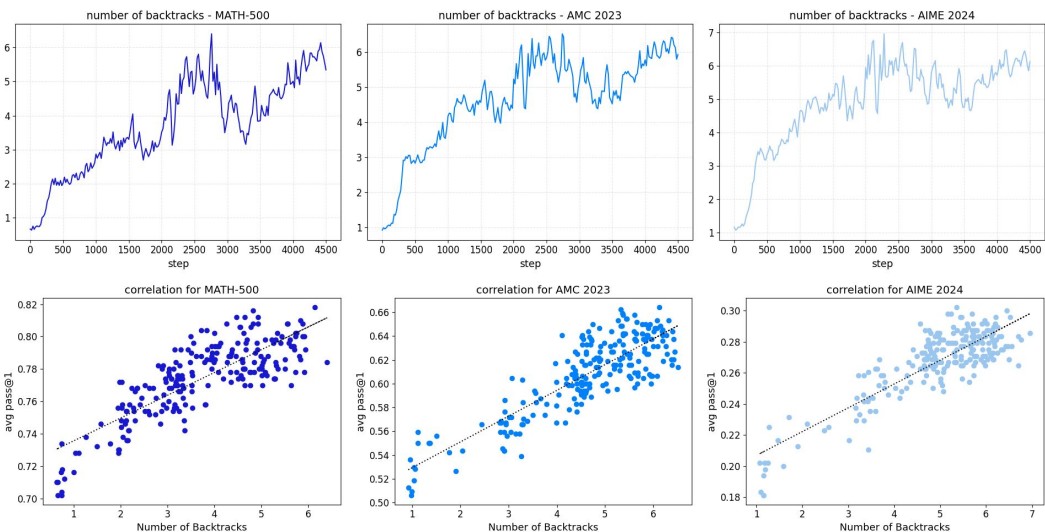

Figure 5: **(Top)** Average number of backtracks performed by the policy during RL training. Our policy learns to backtrack more during RL training. **(Bottom)** Correlation between the number of backtracks and the evaluation performance, computed over all RL checkpoints. The scatter plots demonstrate a positive correlation (Pearson's coefficients – 0.816, 0.851, 0.854) between the number of backtracks performed at test time and the resulting evaluation metrics on our benchmarks.

## 4.2 MAIN RESULTS

Table 1 presents the results of applying ASTRO to `llama-3.1-70b-instruct`. Our SFT checkpoint, named **Llama-3.1-70B-Astro-SFT**, improves over `llama-3.1-70b-instruct` across all benchmarks. Meanwhile, our `llama-3.1-70b-instruct` trained end-to-end via ASTRO with RL, which we name **Llama-3.1-70B-Astro-RL**, outperforms `llama-3.3-70b-instruct` as well as SPOC and Step-KTO applied to `llama-3.1-70b-instruct` while outperforming and matching SPOC and Step-KTO applied to `llama-3.3-70b-instruct` on MATH-500 and AIME 2024, respectively.

Figure 3 visualizes how the generation length of **Llama-3.1-70B-Astro-RL**, as well as reward scores assigned to the model, evolve while running RL. We observe that the policy's CoT generation length initially increases quite noticeably, similar to training runs for other reasoning models (DeepSeek-AI, 2025; Kimi, 2025), and then grows slowly after this initial phase. The reward scores increase monotonically overall, indicating that the policy learns to solve more math problems correctly. Meanwhile, Figure 4 visualizes how the performance of **Llama-3.1-70B-Astro-RL** on MATH-500, AMC 2023 and AIME 2024 evolves during RL. We observe that the evaluation metrics continue to increase steadily across all three benchmarks with more RL. Refer to Appendix E for examples of chain-of-thoughts generated by models trained with ASTRO after SFT and RL.

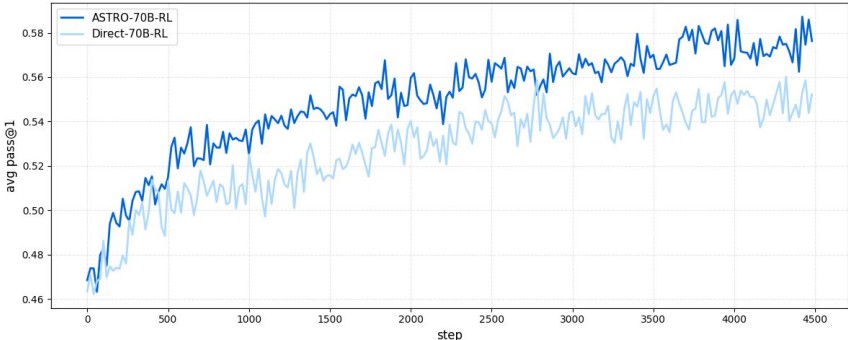

Figure 6: RL training curves for our no-search baseline vs. ASTRO, averaged over the three evaluation benchmarks. The difference between the policy trained with search priors (ASTRO, **dark blue**) and without search priors (Direct, light blue) showcases the importance of infusing search priors for RL.

| Checkpoint | MATH-500 | AMC 2023 | | AIME 2024 | |
| --- | --- | --- | --- | --- | --- |
| | pass@1 | pass@1 | maj@8 | pass@1 | maj@8 |
| Llama-3.1-70B-Direct-RL (best) | 79.8 | 60.5 | 67.8 | 27.1 | 30.3 |
| **Llama-3.1-70B-ASTRO-RL** (best) | **81.8** | **64.4** | **68.8** | **30.0** | **32.0** |

Table 2: Results of training on solutions *without* search priors (Direct) vs. *with* search priors (ASTRO) and running RL, from the same input problems and solutions curated from the same search trees.

### 4.3 IMPACT OF THE SEARCH PRIOR

The reasoning capabilities of ASTRO-trained models can be attributed to both (1) the self-reflection and backtracking priors infused during ASTRO, and (2) the high-quality chain-of-thoughts curated from the search trees (Section 2.2). In this section we verify the importance of the search prior, a core component of ASTRO that shapes the policy's behavior during SFT and RL.

**Search behavior during RL.** We seek to better understand the role of ASTRO's search priors. Figure 5 **(Top)** shows that our policy learns to perform more self-reflections and backtracks during RL, while Figure 5 **(Bottom)** shows that the number of backtracks employed at test-time is highly correlated with evaluation metrics, with the Pearson's coefficient scoring 0.816, 0.851 and 0.854 across the benchmarks. Refer to Appendix C.5 for a similar analysis on the generation length.

**Training without search priors.** To isolate the impact of search priors on ASTRO, we finetune `llama-3.1-70b-instruct` on the same problem set used for **Llama-3.1-70B-ASTRO-SFT** along with the solutions curated from the same search trees, but only using *direct* solutions that arrive at the correct answer without any search behavior. We name the resulting checkpoint **Llama-3.1-70B-Direct-SFT**. Afterwards, we use the same set of prompts used for training **Llama-3.1-70B-ASTRO-RL** and perform RL on **Llama-3.1-70B-Direct-SFT** for the same number of training steps to obtain **Llama-3.1-70B-Direct-RL**. Refer to Appendix D for more details on this experiment.

Figure 6 compares the averaged evaluation metrics over the entire RL training for both models, and shows that our policy with search priors (ASTRO) exhibit better training efficacy and upper bound compared to our policy without search priors (Direct). Table 2 shows the evaluation metrics of the best checkpoints. While RL also allows **Llama-3.1-70B-Direct-RL** to achieve gains across our benchmarks, its best checkpoint underperforms **Llama-3.1-70B-ASTRO-RL** across all benchmarks by nontrivial margins. Our results demonstrate the importance of the search priors used in ASTRO.

**Qualitative Example.** Refer to Figure 7 for an example output from **Llama-3.1-70B-ASTRO-RL**. Our model reflects over its own solution, backtracks to a previous reasoning step and repeats this process, exploring over the solution space until arriving at a final answer. Every CoT generated by our model trained with ASTRO can be mapped to a directed graph with each node representing a discrete reasoning step, allowing for easier visualization and tracking of the model's reasoning. Refer to Appendix E.2 for more examples of search trajectories generated by **Llama-3.1-70B-ASTRO-RL**.

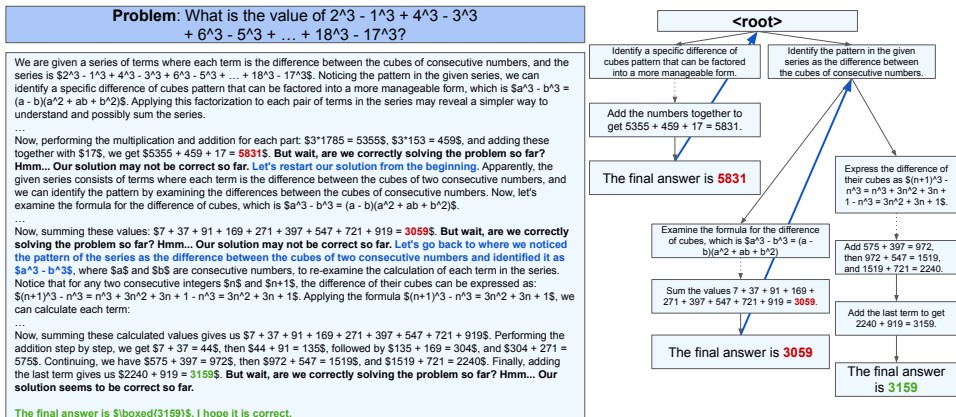

Figure 7: Example of LLAMA-3.1-70B-ASTRO-RL solving a problem from AMC 2023. **(Left)** The model generates a long CoT that contains self-reflections (**black**) and backtracking (**blue**) and explores the solution space, backtracking from reasoning traces with the incorrect answer (**red**) until arriving at a reasoning trace with the correct answer (**green**). **(Right)** The CoT is mapped to a directed graph whose vertices represent discrete reasoning steps and edges represent the search procedure.

## 5 RELATED WORK

**Self-correction for reasoning.** Setlur et al. (2024) find that training models on per-step negative responses help to unlearn spurious correlations in the positive data. Similarly, Ye et al. (2025) show that adding erroneous solution steps followed by their corrections to pretraining data improve math performances. Qu et al. (2024) train LMs to improve their math solutions in-context via iterative fine-tuning, and Kumar et al. (2025) show that training LMs to perform self-correction via multi-turn RL improve their math reasoning abilities. Snell et al. (2024) report that scaling the number of revisions performed in-context for math problems improves the accuracies of the final answers.

**Learning to search for reasoning.** Laskin et al. (2022) propose algorithm distillation, which distills RL algorithms into neural networks in an autoregressive manner by training on their learning histories. Also, Yang et al. (2022) introduce procedure cloning, which learns from series of expert computations and generalizes better to unseen environments. Lehnert et al. (2024) introduce Searchformer, which predicts the sequences of the A* search algorithm while solving the Sokoban puzzle. Meanwhile, Gandhi et al. (2024) train LMs to imitate different search strategies for the Countdown task. Xiang et al. (2025) propose applying procedure cloning to LLM search traces to improve their math reasoning abilities, showcasing several proof-of-concept examples.

**Reinforcement learning for reasoning.** Luo et al. (2025); Wang et al. (2024); Shao et al. (2024) show that training language models via RL with reward models improves their math reasoning abilities, and Yang et al. (2024b); Lambert et al. (2024) propose using RL with verifiable rewards. DeepSeek-AI (2025); Kimi (2025) use RL to train LLMs to generate long CoTs and greatly improves their reasoning capabilities. More recently, Hu et al. (2025); Yu et al. (2025); Liu et al. (2025) perform RL on Qwen models (Yang et al., 2024a;b) and achieve significant performance improvements.

## 6 CONCLUSION

We introduce ASTRO, a new framework for training LLMs to reason like search algorithms. ASTRO operates in three stages: (1) data generation (2) supervised fine-tuning, and (3) reinforcement learning. We first build search trees using Monte Carlo Tree Search (MCTS), linearize the trees into node sequences, and translate them into long CoTs that integrate self-reflection and backtracking. Then, we perform SFT on the long CoTs to instill the search priors, and further improve reasoning capabilities via RL. Our model trained using ASTRO performs in-context search and solves challenging math problems that require iterative revisions. Moreover, our model learns to reflect and backtrack more frequently during RL, and we verify the importance of the search priors used in ASTRO. Our framework offers a systematic recipe to instill robust reasoning capabilities into open-source LLMs.

ETHICS STATEMENT

Our work does not involve human subjects or sensitive data, and solely uses open-source models and datasets to train language models to solve challenging math problems via in-context search. We believe there is no ethical concern to be raised regarding our work.

REPRODUCIBILITY STATEMENT

Reproducing ASTRO takes three distinctive stages: (1) data generation (Section 2), (2) supervised fine-tuning (SFT, Section 3.1) and (3) reinforcement learning (RL, Section 3.2).

For the data generation stage, we describe Monte Carlo Tree Search (MCTS), including its hyper-parameters in Section 2.2 and equations for its implementation in Section A.1. We fully describe our search tree linearization algorithm in Algorithm 1 in Section 2.3, and we provide our few-shot prompts for converting the node sequences into long CoT in Section A.3. Moreover, we share details of our SFT in Section B ( B.1 and B.2), and RL in Section C ( C.1 and C.2).

While we are unable to share code for legal reasons during this submission, in our attachment we share the dataset used for the SFT and RL stages such that any off-the-shelf library can be used to finetune open-weight models to perform in-context search.

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

# Appendix

## A  DATA GENERATION DETAILS

### A.1  MONTE CARLO TREE SEARCH

We provide mathematical descriptions of our Monte Carlo Tree Search (MCTS) used to create the initial search trees from which we curate our search trajectories, as described in Section 2.2.

**Selection.** We use the Predictor+Upper Confidence bounds applied to Trees (PUCT, Silver et al. (2016)) to balance exploration and exploitation. From any state $S_t$, given the action index $i \in [1...k]$, the score of taking action $a_i$ from state $S_t - Q(S_t, a_i)$, the total visit count of $S_t - N(S_t)$, and the visit count of taking action $a_i$ from $S_t - N(S_t, a_i)$, we perform selection as:

$$S_{t+1}^* = \underset{(S_{t+1} = S_t \to a_i)}{\arg\max} \left[ Q(S_t, a_i) + c_{\text{puct}} \cdot \Pi_{\text{LM}}(a_i | S_t) \frac{\sqrt{N(S_t)}}{1 + N(S_t, a_i)} \right] \quad (2)$$

**Expansion.** $\Pi_{\text{LM}}$ takes $\mathbf{x}$ and $(s_0, \cdots, s_t)$ as the input from state $S_t$, and first samples $k$ next actions. For each action, we sample $M$ rollouts and score them using $\mathcal{V}$. Then, we average the rollout scores for each new action $a_i$ ($i \in [1...k]$) to compute the reward scores for the new states, as the following:

$$R(S_{t+1}) = \frac{1}{M} \sum_{j \in [1...M]} \mathcal{V}(\Pi_{\text{LM},j}(S_{t+1})) \quad (3)$$

**Backpropagation.** We backpropagate the reward scores from Eq. 3 to update the Q-values of the search tree. During this process, we update the number of visits of each node – $N(s_t)$, and the Q-value of each (state, action) pair – $Q(s_t, a)$, using the Q-values and visit counts of the children nodes of $S_{t+1} = (S_t, a)$, along with the reward score $R(S_{t+1})$, as the following:

$$N(s_t) = N(s_t) + 1 \quad (4)$$

$$Q(S_t, a) = \frac{\sum_{i=1}^{K} Q(S_{t+1}, a_i) \cdot N(S_{t+1}, a_i) + R(S_{t+1})}{\sum_{i=1}^{K} N(S_{t+1}, a_i) + 1} \quad (5)$$

### A.2  SELF-EVALUATION

We provide more details regarding the self-evaluation used during linearization (Section 2.3) of our search trees. We prompt the policy $\Pi_{\text{LM}}$ to provide a reason for whether the given solution contains valid reasoning or not by first "thinking" whether the solution skips complicated math operations, overlooks important casework or performs a lucky guess to arrive at the answer. Then, the policy returns a binary answer of "yes" or "no" to conclude whether the reasoning is valid or not. We use a few-shot prompt, presented below, to instruct the policy to follow the exact demonstrated format for returning validation thoughts followed by a "yes" or "no" answer. Moreover, we generate $N = 8$ self-evaluations per solution and employ a form of self-consistency (Wang et al., 2023) to obtain multiple votes on the quality of the solution to increase the reliability of our self-evaluation, and we obtain an averaged quality score $s_i$ for the solution $S_i$.

$$s_i = \frac{1}{N} \sum_{j=1}^{N} \mathbb{1} \left[ \Pi_{\text{LM}}(S_i) = \text{yes} \right]$$

We only consider solutions with score $s_i = 1$, which imply unanimous agreement between the $N = 8$ evaluations, to be "high-quality" solutions. We present our few-shot self-evaluation prompt below.

```
Given the following math problem and the solution, decide whether the
    final answer to the problem was obtained via correct reasoning, or
    if the final answer was obtained with insufficient reasoning or a
    lucky guess. Think before arriving at the final answer, which must
    be answered "yes" for correct reasoning, and "no" for incorrect
    reasoning.
```

Examples of answers obtained via incorrect reasoning include:
- The solution presents a complicated mathematical operation (e.g.,
    numerical methods for integration) that must be completed to solve
    the problem and returns the answer without any details on how the
    operation was performed to arrive at the answer.
- The solution presents casework for arriving at the final answer (case
    1, case 2, ...) but returns the final answer after considering only
    a subset of the cases without explaining why the other cases were
    not considered.
- The solution performs a random guess for the final answer without any
    justified explanation for why that guess was made. Educated guesses
    are okay.

Below are a few examples of math problems and their solutions, the
    thinking process for determining why the reasoning is correct or
    incorrect, and the final decision for whether the reasonign is
    correct or not.
---
Problem: Let f(x)=|x-p|+|x-15|+|x-p-15| , where 0 < p < 15 . Determine
    the minimum value taken by f(x) for x in the interval p \\leq
    x\\leq15.

Solution:
## Step 1: Understanding the given function
The given function is f(x) = |x-p| + |x-15| + |x-p-15|, where 0 < p <
    15. We need to find the minimum value of f(x) for x in the interval
    p \\leq x \\leq 15.

## Step 2: Analyzing the behavior of f(x) in the interval p \\leq x
    \\leq 15
For x in the interval p \\leq x \\leq 15, we observe the behavior of
    each absolute value term. Since x \\geq p, |x-p| = x-p. For |x-15|,
    since x \\leq 15, |x-15| = 15-x. The term |x-p-15| requires closer
    examination because its behavior changes based on the value of x
    relative to p+15. However, given that x \\leq 15 and p < 15, x will
    always be less than or equal to 15, and thus |x-p-15| = p+15-x for x
    in the given interval.

## Step 3: Simplifying the function for the given interval
Substituting the simplified expressions back into f(x) gives us f(x) =
    (x-p) + (15-x) + (p+15-x). Simplifying this, f(x) = x - p + 15 - x +
    p + 15 - x = 30 - x.

## Step 4: Finding the minimum value of f(x) in the interval
The function f(x) = 30 - x is a linear function with a negative slope,
    meaning its minimum value in the interval p \\leq x \\leq 15 will
    occur at the maximum value of x in that interval, which is x = 15.
    Substituting x = 15 into f(x), we find f(15) = 30 - 15 = 15.

The final answer is: \\boxed{15}.

Correctness thoughts: The solution correctly considers the range of p,
    given as p \\leq x\\leq15, and uses this information to remove the
    absolute value operators and simplify the given function. Then, it
    correctly reasons that the function f(x) = 30 - x is decreasing in
    the interval p \\leq x\\leq15 and therefore the minimum value must
    happen at x = 15, which is 30 - 15 = 15.

Is correct reasoning: yes
---
Problem: A machine-shop cutting tool has the shape of a notched circle,
    as shown. The radius of the circle is \\sqrt{50} cm, the length of
    AB is 6 cm and that of BC is 2 cm. The angle ABC is a right angle.
    Find the square of the distance (in centimeters) from B to the
    center of the circle. [asy] size(150);

```
    defaultpen(linewidth(0.6)+fontsize(11)); real r=10; pair O=(0,0),
    A=r*dir(45),B=(A.x,A.y-r); path P=circle(O,r); pair
    C=intersectionpoint(B--(B.x+r,B.y),P); // Drawing arc instead of
    full circle //draw(P); draw(arc(O, r, degrees(A), degrees(C)));
    draw(C--B--A--B); dot(A); dot(B); dot(C); label("A",A,NE);
    label("B",B,S); label("C",C,SE); [/asy]

Solution:
## Step 1: To solve this problem, we need to find the square of the
    distance from point B to the center of the circle.
The problem involves a notched circle with a radius of \\sqrt{50} cm and
    points A, B, and C as described.

## Step 2: Let's denote the center of the circle as O and the foot of
    the altitude from O to AB as D.
We need to find the square of the distance from B to the center of the
    circle, which is OB^2.

## Step 3: Using the Pythagorean theorem, we can express the
    relationship between the lengths of the sides of triangle OAB.
By applying the Pythagorean theorem to triangle OAB, we get OA^2 = OB^2
    + AB^2. Substituting the given values, we have 50 = OB^2 + 36.

## Step 4: Similarly, for triangle OBC, we can use the Pythagorean
    theorem to relate the lengths of its sides.
Applying the Pythagorean theorem to triangle OBC, we get OC^2 = OB^2 +
    BC^2. Substituting the given values, we have 50 = OB^2 + 4.

## Step 5: By solving the equations from steps 3 and 4, we can find the
    value of OB^2.
From step 4, we can express OB^2 as 50 - 4 = 46. This value represents
    the square of the distance from point B to the center of the circle.

The final answer is: \\boxed{26}.

Correctness thoughts: The problem is asking for OB^2 according to our
    notation, and Step 5 indicates that OB^2 is equal to 46. However,
    the final answer is 26 without any explanation as to why it returned
    26 instead of 46.

Is correct reasoning: no
---
Based on the examples above that demonstrate how to determine the
    correctness of a solution to a given math problem, determine the
    correctness of the solution to the math problem given below.
Make sure to end your thoughts with a newline followed by "Is correct
    reasoning: yes" or "Is correct reasoning: no".

Problem: %s

Solution: %s

Correctness thoughts:
```

## A.3 CHAIN-OF-THOUGHT GENERATION

We provide more details for how we convert the linearized sequence of nodes into a natural language CoT that integrates self-reflection and backtracking (Section 2.4). As we iterate over the sequence of nodes, for each node $n_t$ at timestep $t$ we sample $\Pi_{\text{LM}}$ with few-shot prompts conditioned upon the CoT generated so far ($\mathbf{y}_{1...t-1}$), where (a) case $1 - n_t$ is a descendant of $n_{t-1}$: the few-shot prompt involves directly rewriting $s_t$ into a CoT format, and (b) case $2 - n_t$ is an ancestor of $n_{t-1}$: the few-shot prompt involves writing a phrase the backtracks from $n_{t-1}$ to $n_t$. Below we present the few-shot prompts for rewriting and backtracking.

Few-shot prompt for rewriting:

```
Given a partially thought-out solution to a math problem so far and the
    current step for solving the problem, your job is to rewrite the
    *current step* into a thought that smoothly continues the previous
    thoughts. This rewritten thought should only address the contents of
    the current step itself, nothing more or less. Make sure to follow
    the requirements below:
- Remember that your job is to simply rewrite the current solution step
    into a thought process.
- Write the solution thought such that it continues smoothly from the
    previous solution thoughts.
- Include all of the reasoning provided in the current solution step in
    the thought process.
- Do NOT repeat anything from what is written in the previous solution
    thoughts.
- Do NOT think about the problem further than what is provided in the
    current solution step.

Below are examples of how to rewrite the current solution step into a
    thought process. Make sure to follow the format taken by the
    examples below.
---
Previous solution step: None

Current solution step:
## Define the variables and set up an equation.
Let's define our variables: B = number of students with blond hair, E =
    number of students with blue eyes. We're given that the total number
    of students is 30, so B + E - 6 + 3 = 30, because we need to
    subtract the 6 students counted twice for having both blond hair and
    blue eyes and add the 3 students who have neither.

Rewritten solution thoughts: Let's begin by denoting B as the number of
    students with blond hair and E as the number of students with blue
    eyes. We're given that there are 30 students in total, so B + E - 6
    + 3 = 30, because we need to subtract the six students counted twice
    for having both blond hair and blue eyes and add the three students
    who have neither.
---
Previous solution thoughts: Let's denote B as the number of students
    with blond hair and E as the number of students with blue eyes.
    Let's recall the formula for the union of sets is \\|A \\cup B\\| =
    \\|A\\| + \\|B\\| - \\|A \\cap B\\|. We're given that there are 30
    students in total, so B + E - 6 + 3 = 30, because we need to
    subtract the six students counted twice for having both blond hair
    and blue eyes and add the three students who have neither. We're
    also given that there are twice as many students with blond hair as
    with blue eyes, so B = 2E.

Current solution step:
## Solve the equation
Substituting B = 2E into the equation gives us: 2E + E - 6 + 3 = 30.

Rewritten solution thoughts: Let's substitute this relationship into the
    previous equation to get: 2E + E - 6 + 3 = 30.
---
Previous solution thoughts: Let's first recall the formula for the
    volume of a sphere. For radius r and volume V, we know that V =
    \\frac{4}{3}\\pi r^3. From the problem, we know that the volume of
    the larger sphere is 288\\pi cubic units. So, we can set up an
    equation \\frac{4}{3}\\pi r_1^3 = 288\\pi, where r_1 is the radius
    of the larger sphere. Solving for r_1 gives r_1^3 = \\frac{288\\pi
    \\cdot 3}{4\\pi} = 216 and r_1 = 6 units. Meanwhile, we are given
    that the volume of the smaller sphere is 12.5 percent of the larger
```

```
    sphere's volume. Given that 12.5 percent = 0.125, we know that the
    volume of the smaller sphere is V_2 = 0.125 \\cdot 288\\pi = 36\\pi
    cubic units.

Current solution step:
## Set up the equation for the smaller sphere and find its radius.
Setting up the volume equation for the smaller sphere, \\frac{4}{3}\\pi
    r_2^3 = 36\\pi, where r_2 is the radius of the smaller sphere.
    Solving for r_2 gives r_2^3 = \\frac{36\\pi \\cdot 3}{4\\pi} = 27
    and r_2 = 3 units.

Rewritten solution thoughts: Let's set up the equation for the volume of
    the smaller sphere with r_2 being the radius of the smaller sphere:
    \\frac{4}{3}\\pi r_2^3 = 36\\pi. Then, let's solve for r_2 and
    obtain r_2^3 = \\frac{36\\pi \\cdot 3}{4\\pi} = 27 and r_2 = 3 units.
---
Previous solution thoughts: Let's first consider the overall strategy
    for solving the problem - we need to first rewrite the given
    equation in standard form to find the distance between the vertices.
    To this end, we should complete the square for the x and y terms.

Current solution step:
## Complete the square for the x term.
Let's start with the x term. We can factor out the coefficient of x^2
    and then complete the square by adding and subtracting the square of
    half the coefficient of x. The coefficient of x^2 is 9, so factoring
    that out gives us 9(x^2 + 6x). To complete the square, we add and
    subtract (6/2)^2 = 9 inside the parentheses, which gives us 9(x^2 +
    6x + 9 - 9) = 9(x^2 + 6x + 9) - 81.

Rewritten solution thoughts: Let's start with the x term. We can factor
    out the coefficient of x^2 and then complete the square by adding
    and subtracting the square of half the coefficient of x. The
    coefficient of x^2 is 9, so factoring that out gives us 9(x^2 + 6x).
    To complete the square, let's add and subtract (6/2)^2 = 9 inside
    the parentheses, so that we have 9(x^2 + 6x + 9 - 9) = 9(x^2 + 6x +
    9) - 81.
---
Based on the examples above that demonstrate how to rewrite a solution
    step into solution thoughts, rewrite the solution step given below
    into solution thoughts. This rewritten thought should only address
    the contents of the current step itself, nothing more or less.
    Again, make sure to follow the requirements below:
- Remember that your job is to simply rewrite the current solution step
    into a thought process.
- Write the solution thought such that it continues smoothly from the
    previous solution thoughts.
- Include all of the reasoning provided in the current solution step in
    the thought process. Do NOT miss any single line provided in the
    solution step, and especially pay careful attention to equations
    spanning multiple lines of the solution.
- Do NOT repeat anything from what is written in the previous solution
    thoughts.
- Do NOT think about the problem further than what is provided in the
    current solution step. To repeat, NEVER include thoughts that are
    unsupported in the current solution step.

Previous solution thoughts: %s

Current solution step:
%s

Rewritten solution thoughts:
```

Few-shot prompt for backtracking:

```
Given a partially thought-out solution to a math problem so far which
    ends by identifying itself to be incorrect and needing to backtrack,
    the "backtracked" step that the solution is supposed to backtrack
    to, your job is to continue the existing solution thoughts by
    backtracking to the part of the solution that corresponds to the
    "backtracked step". Make sure to follow the requirements below:
- Make sure that the solution continuation only addresses the backtrack
    step and nothing more or less.
- Note that if the backtracking requires a restart of the solution, the
    "backtrack step" is given as "RESTART". In this case, do NOT begin
    your response with "RESTART". Rather, use natural phrases like
    "Let's restart from the beginning..." or "Let's start over..." to
    indicate that the solution must be restarted.
- Do not use phrases such as "Let's restart" or "Let's start over" if
    the backtracked step is not "RESTART", but use a phrase such as
    "Let's backtrack to..." or "Let's go back to...".

Below are examples of how to backtrack to a previous step, continue to
    the current step and continue the solution. Make sure to follow the
    format taken by the examples below.
---
Previous solution thoughts: Let's begin by defining B as the number of
    students with blond hair and E as the number of students with blue
    eyes. If we recall correctly, the formula for the union of sets is
    \\|A \\cup B\\| = \\|A\\| + \\|B\\| - \\|A \\cap B\\|. Since there
    are 30 students in total, B + E - 6 = 30, because we need to
    subtract the six students who have both blond hair and blue eyes.
    But wait, are we correctly solving the problem so far?

Backtrack step: RESTART

Backtrack solution thoughts: Let's restart our solution from the
    beginning.
---
Previous solution thoughts: First, we should recall that the formula for
    the volume of a sphere is V = \\frac{4}{3}\\pi r^3 for radius r and
    volume V. The problem also tells us that the volume of the larger
    sphere is 288\\pi cubic units. Given this information, let's set up
    an equation \\frac{4}{3}\\pi r_1^3 = 288\\pi, with r_1 being the
    radius of the larger sphere. This means that r_1^3 = \\frac{288\\pi
    \\cdot 3}{4\\pi} = 216 and r_1 = 6 units. Meanwhile, we also know
    that the volume of the smaller sphere is 12.5 percent of the larger
    sphere. This means that the volume of the smaller sphere is V_2 =
    0.125 \\cdot 288\\pi = 36\\pi cubic units. Let us now define r_2 as
    the radius of the smaller sphere. Then, \\frac{4}{3}\\pi r_2^3 =
    36\\pi. We can solve this equation for r_2 and find that r_2^3 =
    \\frac{36\\pi \\cdot 3}{4\\pi} = 64 and r_2 = 4 units. But hold on,
    could it be possible that we made a mistake in our solution?

Backtrack step: ## Compute the volume of the smaller sphere.
The volume of the smaller sphere is 12.5 percent of the larger sphere's
    volume. Given that 12.5 percent = 0.125, the volume of the smaller
    sphere is V_2 = 0.125 \\cdot 288\\pi = 36\\pi cubic units.

Backtrack solution thoughts: Let's go back to where we computed the
    volume of the smaller sphere as 36\\pi cubic units.
---
Previous solution thoughts: Let's first consider the overall strategy
    for solving the problem - we need to first rewrite the given
    equation in standard form to find the distance between the vertices.
    To this end, we should complete the square for the x and y terms.
    Let's start with the x term. We can factor out the coefficient of
    x^2 and then complete the square by adding the square of half the
```

```
coefficient of x. The coefficient of x^2 is 9, so factoring that out
    gives us 9(x^2 + 6x). To complete the square, let's add (6/2)^2 = 9
    inside the parentheses and subtract by the same amount outside, so
    that we have 9(x^2 + 6x + 9) - 9 \\cdot 9 = 9(x^2 + 6x + 9) - 81.
    Now let's move on to the y term. We can factor out the coefficient
    of y^2, which is -1, and then complete the square by adding the
    square of half the coefficient of y. The coefficient of y is 10, so
    factoring out -1 gives us -(y^2 - 10y). Let's add (10/2)^2 = 25
    inside the parentheses and subtract by the same amount outside, so
    that we have -(y^2 - 10y + 25) - 25. But wait, are we sure that our
    solution is correct so far?

Backtrack step: ## Complete the square for the x term.
Let's start with the x term. We can factor out the coefficient of x^2
    and then complete the square by adding and subtracting the square of
    half the coefficient of x. The coefficient of x^2 is 9, so factoring
    that out gives us 9(x^2 + 6x). To complete the square, we add and
    subtract (6/2)^2 = 9 inside the parentheses, which gives us 9(x^2 +
    6x + 9 - 9) = 9(x^2 + 6x + 9) - 81.

Backtrack solution thoughts: Let's backtrack to where we completed the
    square for the x term and obtained the correct expression for x,
    which is 9(x^2 + 6x + 9) - 81.
---
Based on the examples above that demonstrate how to continue the
    existing solution thoughts via backtracking, continue the solution
    thought by backtracking to the "backtracked step". Again, make sure
    to follow the requirements below:
- Make sure that the solution continuation only addresses the backtrack
    step and nothing more or less.
- Note that if the backtracking requires a restart of the solution, the
    "backtrack step" is given as "RESTART". In this case, do NOT begin
    your response with "RESTART". Rather, use natural phrases like
    "Let's restart from the beginning..." or "Let's start over..." to
    indicate that the solution must be restarted.
- Do not use phrases such as "Let's restart" or "Let's start over" if
    the backtracked step is not "RESTART", but use a phrase such as
    "Let's backtrack to..." or "Let's go back to...".
- Do NOT think about the problem further than what is provided in the
    backtrack step. To repeat, NEVER try to solve the problem further
    than what the backtrack step does.
- Write everything in one line without newlines, and do not use "##"
    expressions to label steps. Do not begin your response with
    non-english characters (e.g., colon, ">") or any introductory phrase
    like "Sure, here's how to backtrack...". Also, do not begin with
    "Backtrack solution thoughts: ...". Just write the backtracking
    using the backtracked step.

Previous solution thoughts: %s

Backtrack step: %s

Backtrack solution thoughts:
```

## B  SUPERVISED FINE-TUNING DETAILS

### B.1  TRAINING DATASET DETAILS

We present details regarding the statistics during the generation of our supervised fine-tuning (SFT) dataset composed of long-CoT search trajectories (Section 3.1). Table 3 presents the composition of our SFT dataset used for training our models. Table 4 presents a detailed breakdown of the statistics during the intermediate stages of our data curation across the three math dataset sources used for instilling search behaviors during the initial stages of ASTRO.

| Dataset | # problems | # trajectories | # backtracks | # restarts |
|---|---|---|---|---|
| MATH-train | 5,838 | 12,536 | 3,817 | 4,191 |
| NuminaMath (AMC/AIME) | 1,599 | 5,758 | 3,268 | 1,341 |
| NuminaMath (AoPS Forum) | 4,702 | 17,773 | 11,608 | 2,636 |

Table 3: SFT dataset composition for our main ASTRO training run. Here, `# problems` refers to the number of unique problems, `# trajectories` refers to the total number of search trajectories, `# backtracks` refers to the number of backtracks to previous reasoning steps, and `# restarts` refers to the number of solution restarts in each training subset.

| Dataset | # trees | # valid trees | # solutions | # direct | # search |
|---|---|---|---|---|---|
| MATH-train | 6,912 | 6,118 | 12,536 | 5,838 | 6,698 |
| NuminaMath (AMC/AIME) | 2,480 | 1,812 | 5,758 | 1,746 | 4,012 |
| NuminaMath (AoPS Forum) | 11,329 | 6,069 | 17,773 | 4,706 | 13,067 |

Table 4: Detailed statistics of our data curation procedure in ASTRO resulting in our main SFT dataset. Here, `# trees` refers to the number of search trees from the initial set of problems, `# valid trees` refers to the number of search trees with at least one high-quality solution ending with the correct answer, `# solutions` refers to the number of long chain-of-thought solutions curated from the valid search trees, `# direct` refers to the number of solutions without self-reflection or backtracking, and `# search` refers to the number of solutions with self-reflection and backtracking.

### B.2 TRAINING HYPERPARAMETERS

We train our model for one epoch only to provide a better initialization for the RL stage. We use the AdamW (Loshchilov & Hutter, 2019) optimizer with an initial learning rate of 3e-6 and a cosine scheduler, and we set the maximum sequence length to be 8,192 tokens.

### B.3 HARDWARE SPECIFICATIONS

We train our 70B models across 8 GPU nodes with each node consisting of 8 NVIDIA H100 GPUs. Under this setup, training for one epoch on our SFT dataset takes approximately 40 minutes.

### B.4 SFT VISUALIZATIONS

We present visualizations of our SFT using the loss curve and the gradient norms in Figure 8.

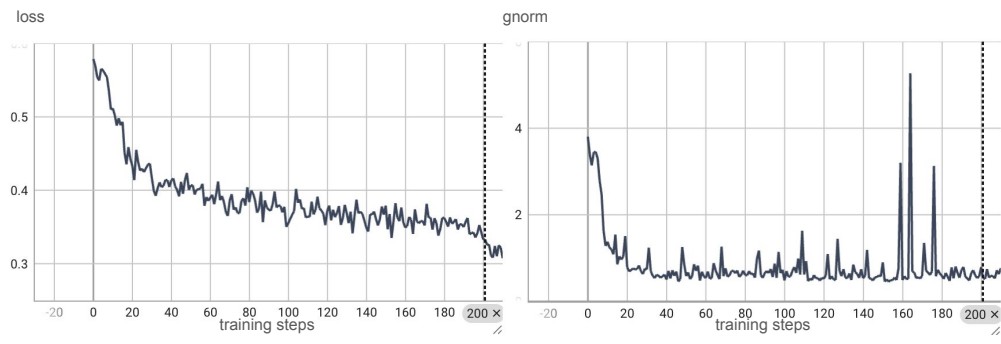

Figure 8: Visualizations for our main SFT run used in ASTRO. **(Left)** Training loss curve. **(Right)** Gradient norms.

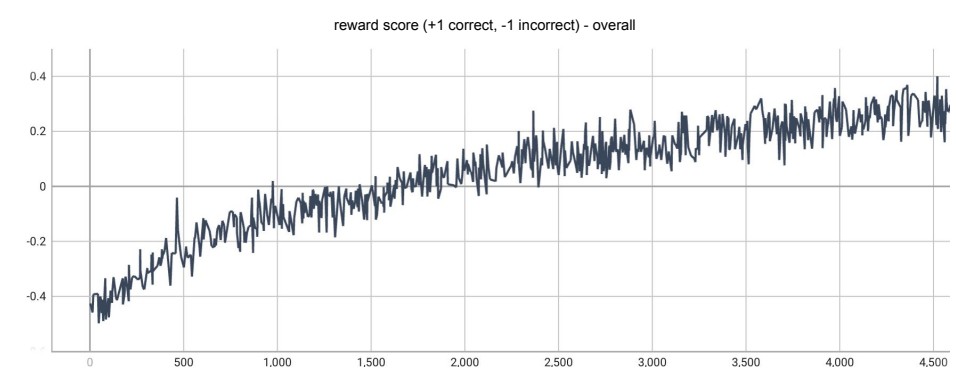

Figure 9: Detailed view of the overall reward scores while training **Llama-3.1-70B-Astro-RL**.

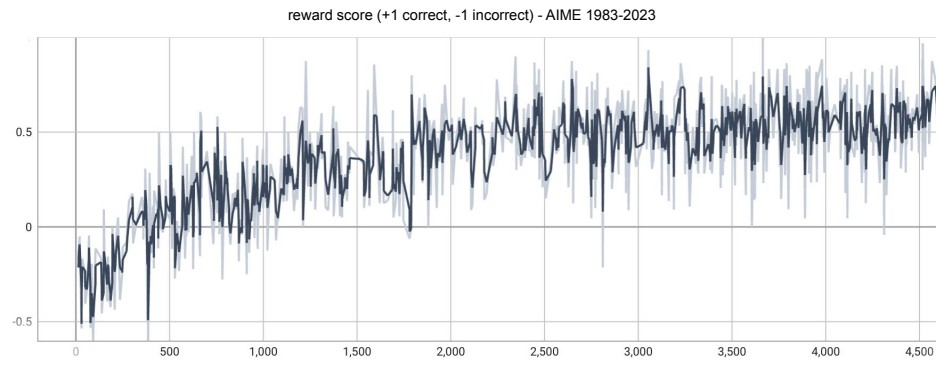

Figure 10: Reward scores on AIME 1983-2023 while training **Llama-3.1-70B-Astro-RL**.

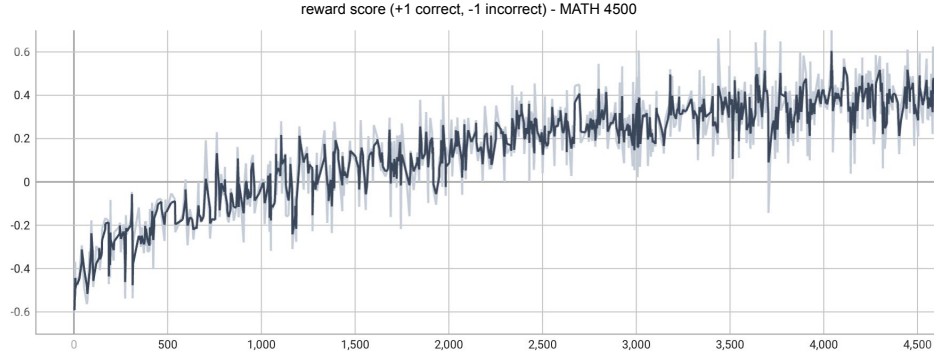

Figure 11: Reward scores on MATH-4500 while training **Llama-3.1-70B-Astro-RL**.

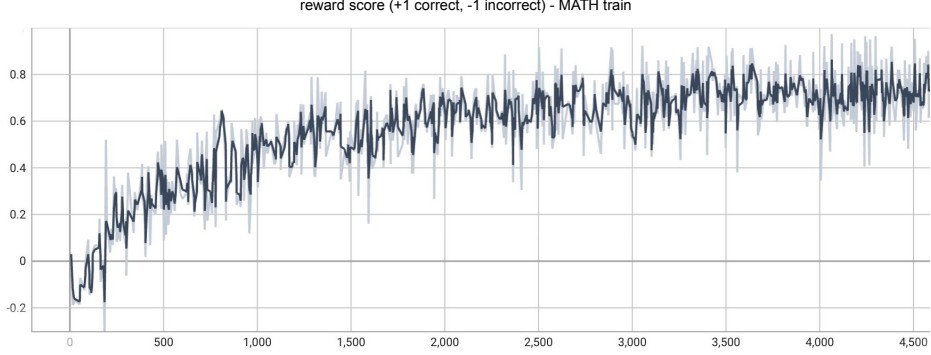

Figure 12: Reward scores on MATH-train while training **Llama-3.1-70B-Astro-RL**.

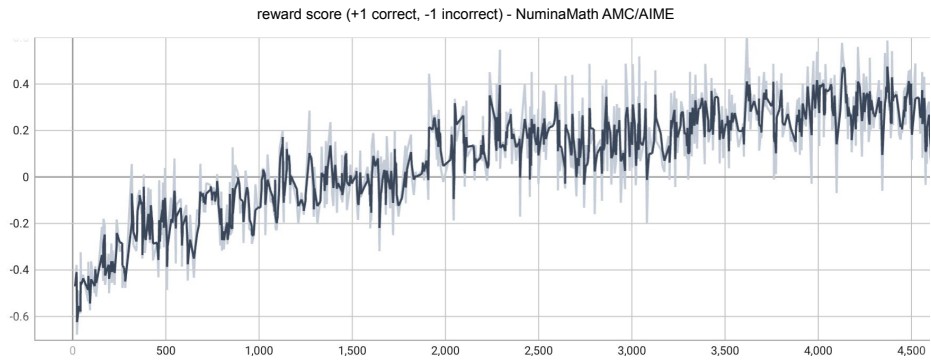

Figure 13: Reward scores on `NuminaMath-AMC/AIME` while training **Llama-3.1-70B-ASTRO-RL**.

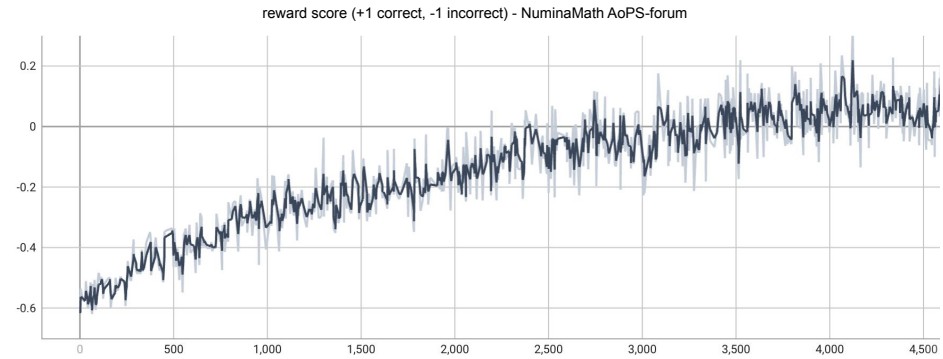

Figure 14: Reward scores on `NuminaMath-AoPS forum` while training **Llama-3.1-70B-ASTRO-RL**.

## C REINFORCEMENT LEARNING DETAILS

### C.1 TRAINING DATASET DETAILS

Table 5 presents detailed statistics for our RL datasets as mentioned in Section 3.2 of the main paper.

### C.2 TRAINING HYPERPARAMETERS

For reinforcement learning (Section 3.2), we use a constant learning rate of 2e-7, four rollouts per prompt for advantage estimation, a batch size of 256 for every step, gradient accumulation step of 1, maximum sequence length of 15,360 tokens, a temperature of 1.0, and 80 warmup steps.

| Dataset | # problems | pass rate avg. | # problems solved | # problems used |
|---|---|---|---|---|
| AIME 1983-2023 | 912 | 40.7% | 721 | 492 |
| MATH-4500 | 4,480 | 72.1% | 4,239 | 1,393 |
| MATH-train | 7,488 | 87.7% | 7,308 | 977 |
| NuminaMath (AMC/AIME) | 3,648 | 49.9% | 2,865 | 1,395 |
| NuminaMath (AoPS Forum) | 8,576 | 27.3% | 5,756 | 4,423 |

Table 5: RL dataset composition for our ASTRO-trained model based on `llama-3.1-70b-instruct`. Here, `# problems total` refers to the total number of problems after string-based filtering for low-quality or unverifiable problems, `pass rate avg.` refers to the mean pass rate across $N = 64$ outputs for all problems, `# problems solved` refers to the number of problems with a non-zero pass rate, and `# problems used` refers to the number of problems falling within the 1% to 75% pass rate range for our SFT policy, which we use for our main RL experiment, in each training subset.

### C.3 HARDWARE SPECIFICATIONS

We run RL with our 70B models across 32 GPU nodes with each node consisting of 8 NVIDIA H100 GPUs. We use 128 GPUs for training and 128 GPUs for inference, respectively. Each of our RL training runs takes about 10 days to complete.

### C.4 REWARD SCORES BY DATASET

We visualize the evolution of reward scores (recall Figure 3) on a per-dataset basis for training **Llama-3.1-70B-ASTRO-RL**, which consist of AIME problems from the years 1983 to 2023, `MATH-4500` which is separated from the MATH-500 evaluation set, along with `MATH-train` and the `AMC/AIME` and `AoPS-forum` subsets of `NuminaMath` (Li et al., 2024). Refer to Figures 9, 10, 11, 12, 13 and 14 for the overall reward scores as well as the reward scores on each training subset, in order.

### C.5 TEST-TIME SCALING ANALYSIS

**Number of generated tokens during evaluation.** In Figure 15, we show the number of tokens generated by **Llama-3.1-70B-ASTRO-RL** during RL. The policy initially generates a steeply increasing number of tokens, and this rate decreases afterwards but keeps a steady pace, similar to the number of backtracks in Figure 4.

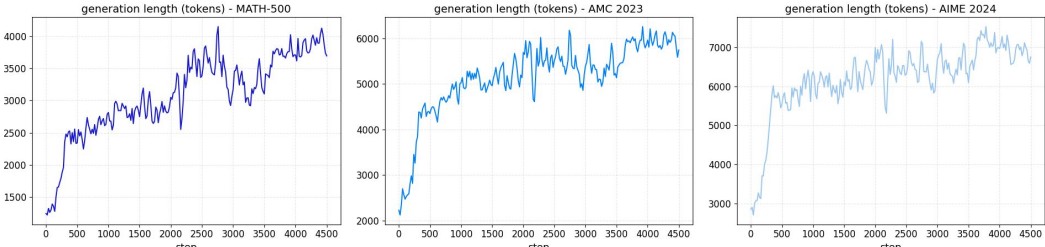

Figure 15: Number of generated tokens on the MATH-500, AMC 2023 and AIME 2024 benchmarks while training **Llama-3.1-70B-ASTRO-RL**. The number of generated tokens initially increases rapidly as the model learns to perform more self-reflections and backtracks (see Figure 4), and then increases more slowly but steadily afterwards.

**Relationship with Evaluation Metrics** We plot the relationship between the number of tokens generated during evaluation and the model's performance. We find a strong correlation with a Pearson's coefficient of 0.858 for MATH-500, 0.836 for AMC 2023 and 0.833 for AIME 2024, showing similar trends to the correlation between the number of backtracks and the model's performance (Section 4.3).

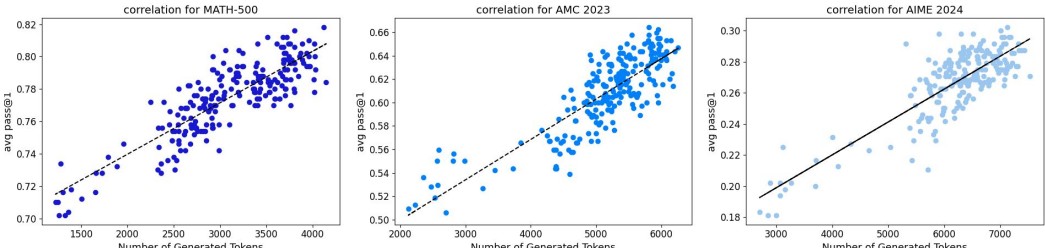

Figure 16: Relationship between the number of generated tokens and evaluation metrics on MATH-500, AMC 2023 and AIME 2024. There is positive correlation between the number of generated tokens and performance on the math benchmarks, indicating that the policy returns more accurate answers as it generates more tokens during inference. The Pearson's coefficients are 0.858 0.836 and 0.833 for MATH-500, AMC 2023 and AIME 2024, respectively.

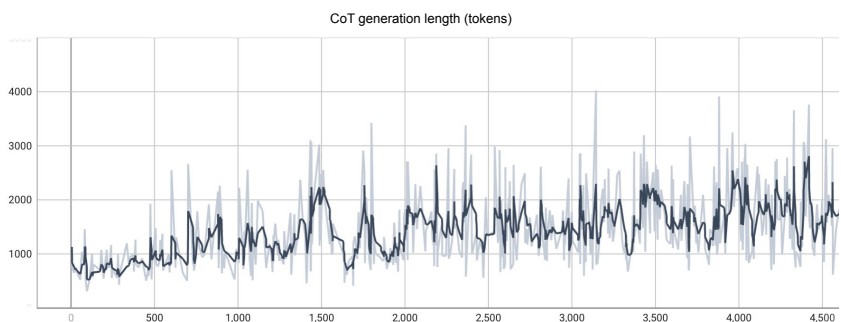

Figure 17: The number of generated tokens while training **Llama-3.1-70B-Direct-RL** via RL.

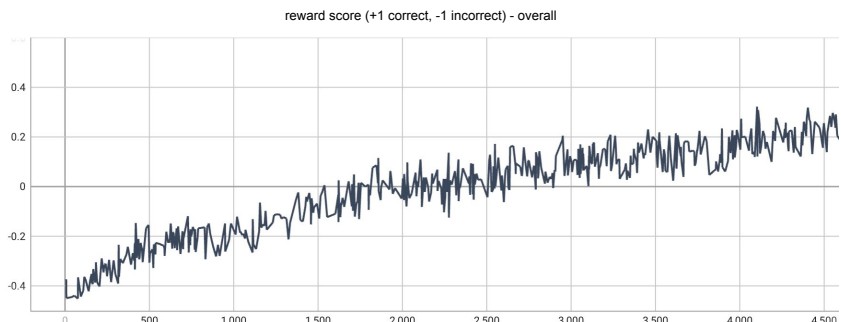

Figure 18: Reward scores measured while training **Llama-3.1-70B-Direct-RL** via RL.

## D DETAILS FOR LLAMA-3.1-70B-DIRECT-RL

We provide more details of **Llama-3.1-70B-Direct-RL** trained according to our experimental setup described in Section 4.3. Note that this model does not have any self-reflection or backtracking priors infused during the SFT stage, as we seek to isolate the effects of the search priors used in ASTRO.

### D.1 TRAINING

Our training setups for the SFT and RL stages are identical to that of training **Llama-3.1-70B-ASTRO-SFT** and **Llama-3.1-70B-ASTRO-RL**, respectively. We focus on the RL portion of training **Llama-3.1-70B-Direct-RL** and share further useful details regarding the model's training.

**CoT generation length.** In Figure 17 we show how the length of the CoT generated by **Llama-3.1-70B-Direct-RL** evolves during RL. Without the self-reflection and backtracking priors, we observe that the number of tokens of the policy's generated CoT increases more slowly, although it does show an increase from about 1K tokens on average to about 2K tokens.

**Reward scores.** In Figure 18 we show how the reward scores of **Llama-3.1-70B-Direct-RL** (+1 for correct, -1 for incorrect) on the training prompts evolve during RL. We observe that its reward scores improve in a similar manner to the reward scores for **Llama-3.1-70B-ASTRO-RL** during RL.

### D.2 EVALUATION

Our evaluation setups for **Llama-3.1-70B-Direct-RL** are identical to those for **Llama-3.1-70B-ASTRO-RL**. We share further useful details regarding the model's evaluations.

**Number of generated tokens.** Figure 19 shows the number of generated tokens for the MATH-500, AMC 2023 and AIME 2024 benchmarks. We observe that the number of tokens generated by **Llama-3.1-70B-Direct-RL** during evaluations increases steadily, albeit at a slower rate without the self-reflection and backtracking priors.

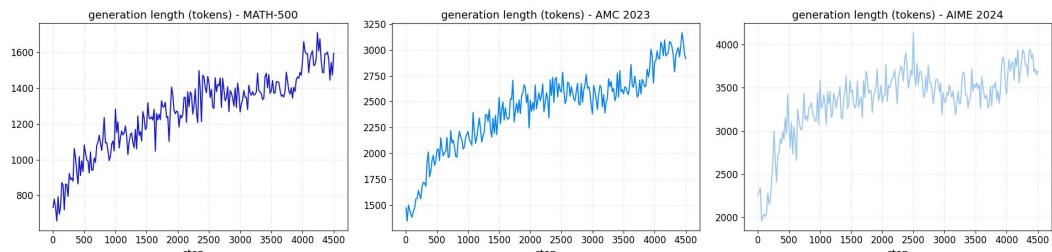

Figure 19: Number of generated tokens on the MATH-500, AMC 2023 and AIME 2024 benchmarks while training **Llama-3.1-70B-Direct-RL**. The number of generated tokens increases steadily with more RL, albeit at a slower rate without the self-reflection and backtracking priors compared to the RL training of **Llama-3.1-70B-ASTRO-RL**.

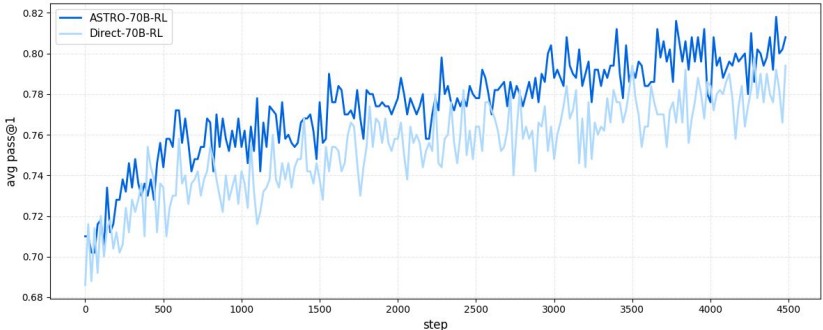

Figure 20: RL training curves for our direct baseline vs. ASTRO on the MATH-500 benchmark.

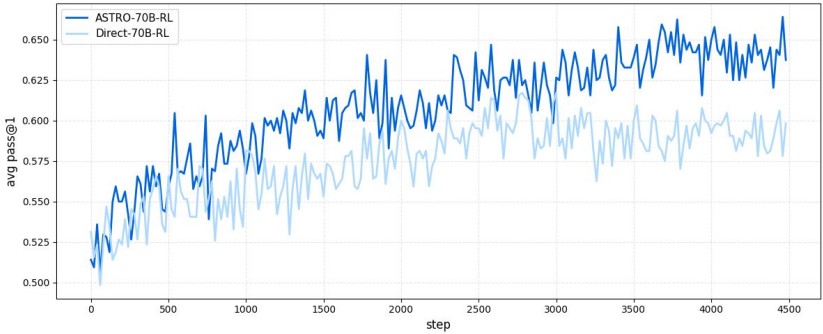

Figure 21: RL training curves for our direct baseline vs. ASTRO on the AMC 2023 benchmark.

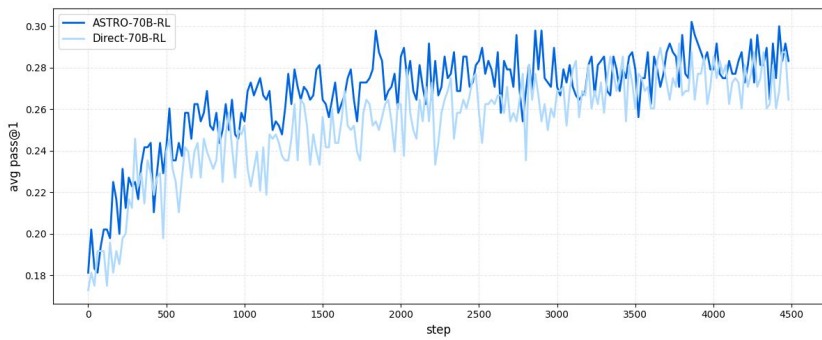

Figure 22: RL training curves for our direct baseline vs. ASTRO on the AIME 2024 benchmark.

**Comparisons with ASTRO.** We share detailed comparisons of the evaluation results between the direct baseline and ASTRO during RL, for each of our three math benchmarks. Refer to Figures 20, 21 and 22 for comparisons on the MATH-500, AMC 2023 and AIME 2024 benchmarks, respectively.

# E QUALITATIVE EXAMPLES

We include qualitative examples of chain-of-thoughts generated by different models trained using ASTRO in our experiments for a better understanding of the model behavior.

## E.1 ASTRO-SFT

**Llama-3.1-70B-ASTRO-SFT** already performs autoregressive search (self-reflections and backtracks), averaging about 1.2 tokens and 0.8 backtracks for MATH-500 and 3K tokens and 1.1 backtracks for AIME 2024. The model explores over a wide solution space, generating both correct and incorrect solution traces until it converges to the final answer and terminates its search process.

```
--------------------------------------------------------------------------
Problem (AIME 2024): Every morning Aya goes for a $9$-kilometer-long
    walk and stops at a coffee shop afterwards. When she walks at a
    constant speed of $s$ kilometers per hour, the walk takes her $4$
    hours, including $t$ minutes spent in the coffee shop. When she
    walks at $s + 2$ kilometers per hour, the walk takes her $2$ hours
    and $24$ minutes, including $t$ minutes spent in the coffee shop.
    Suppose Aya walks at $s + \frac{1}{2}$ kilometers per hour. Find the
    number of minutes the walk takes her, including the $t$ minutes
    spent in the coffee shop.

Correct answer: 204
--------------------------------------------------------------------------
Model response:

Let's start by understanding the problem and defining the variables,
    where Aya walks a 9-kilometer-long walk and then stops at a coffee
    shop, and we need to consider her walking at different speeds,
    denoted as $s$ kilometers per hour, $s + 2$ kilometers per hour, and
    $s + \frac{1}{2}$ kilometers per hour, and the total time for the
    walk and coffee shop visit varies based on her speed, with the time
    spent walking and at the coffee shop being affected by these speed
    changes. First, let's establish the relationships between Aya's
    walking speed, the time spent walking, and the time spent at the
    coffee shop. When Aya walks at $s$ kilometers per hour, the time
    spent walking is $\frac{9}{s}$ hours. The total time for the walk
    and the coffee shop visit is given as $4$ hours, which is equivalent
    to $\frac{9}{s} + \frac{t}{60}$, allowing us to convert $t$ minutes
    into hours. This relationship can be expressed as $\frac{9}{s} +
    \frac{t}{60} = 4$. Considering the different speeds at which Aya
    walks, let's set up an equation for the second scenario where her
    speed is $s + 2$ kilometers per hour. The time spent walking at this
    speed is $\frac{9}{s+2}$ hours. Given that the total time for the
    walk and the coffee shop visit is $2$ hours and $24$ minutes, which
    is equivalent to $\frac{9}{s+2} + \frac{t}{60} = 2 + \frac{24}{60}$,
    we can express this relationship as $\frac{9}{s+2} + \frac{t}{60} =
    2.4$. But wait, are we correctly solving the problem so far?

Hmm... Our solution seems to be correct so far. Now that we have
    established the relationships between Aya's walking speed, the time
    spent walking, and the time spent at the coffee shop, let's utilize
    the two equations $\frac{9}{s} + \frac{t}{60} = 4$ and
    $\frac{9}{s+2} + \frac{t}{60} = 2.4$ to solve for $s$ and $t$. By
    subtracting the second equation from the first, we can eliminate
    $t$, which gives us $\frac{9}{s} - \frac{9}{s+2} = 4 - 2.4$. Upon
    simplifying, we obtain $\frac{9}{s} - \frac{9}{s+2} = 1.6$. Given
    the equation $\frac{9}{s} - \frac{9}{s+2} = 1.6$, we need to combine
```

the fractions on the left side, which leads to $\frac{9(s+2) - 9s}{s(s+2)} = 1.6$. Upon simplifying, we obtain $\frac{18}{s^2 + 2s} = 1.6$. Multiplying both sides by $60$ to clear the fraction yields $9s(s+2) - 18 = 96s + 192$. Expanding the left side results in $9s^2 + 18s - 96s - 192 = 18$. Simplifying further gives $9s^2 - 78s - 210 = 0$. To solve for $s$, we can use the quadratic formula, $s = \frac{-b \pm \sqrt{b^2 - 4ac}}{2a}$, where $a = 9$, $b = -78$, and $c = -210$. Substituting these values in gives $s = \frac{78 \pm \sqrt{(-78)^2 - 4(9)(-210)}}{2(9)}$. This simplifies to $s = \frac{78 \pm \sqrt{6084 + 7560}}{18}$. Thus, $s = \frac{78 \pm \sqrt{13644}}{18}$, which further simplifies to $s = \frac{78 \pm 116.58}{18}$. For positive $s$, we take the positive root, yielding $s = \frac{78 + 116.58}{18}$. Now, let's calculate the numerical value of $s$ by evaluating the expression $\frac{78 + 116.58}{18}$, which simplifies to $\frac{194.58}{18}$, and this further simplifies to $10.81$. But wait, are we correctly solving the problem so far?

Hmm... Our solution seems to be correct so far. Now, let's use the first equation to solve for $t$ with the known value of $s = 10.81$. Substituting $s$ into $\frac{9}{s} + \frac{t}{60} = 4$, we get $\frac{9}{10.81} + \frac{t}{60} = 4$. This means $\frac{t}{60} = 4 - \frac{9}{10.81}$. Let's calculate $\frac{9}{10.81}$, which is approximately $0.832$. Therefore, we have $\frac{t}{60} = 4 - 0.832$. So, $\frac{t}{60} = 3.168$. Then, to find $t$, we multiply both sides by $60$, yielding $t = 3.168 \times 60$. Therefore, $t = 190.08$ minutes. Now, to calculate the total time spent walking and at the coffee shop at the given speed, we recognize that the total time is the sum of the walking time and the time spent at the coffee shop, so let's express this relationship as $Total\ Time = \frac{9}{s + \frac{1}{2}} + t$, where $\frac{9}{s + \frac{1}{2}}$ represents the walking time and $t$ is the time spent at the coffee shop. Now, substitute $s = 10.81$ into the expression $s + \frac{1}{2}$ to obtain $s + \frac{1}{2} = 10.81 + 0.5 = 11.31$. Given the speed $s + \frac{1}{2} = 11.31$, let's calculate the walking time by using the expression $\frac{9}{s + \frac{1}{2}}$, which becomes $\frac{9}{11.31}$. This simplifies to approximately $0.796$ hours. Given the walking time of approximately $0.796$ hours spent at the speed $s + \frac{1}{2} = 11.31$, let's convert this time into minutes by multiplying it by $60$, which gives us $0.796 \times 60$. Noting that we already have the walking time expressed in hours as $0.796$, to find the total time including the coffee shop visit, we need to add the $t = 190.08$ minutes spent at the coffee shop to this walking time. However, since the walking time is given in hours, we should convert it into minutes by multiplying by $60$, resulting in $0.796 \times 60 = 47.76$ minutes, and then add $t$ to get the total time in minutes as $47.76 + 190.08 = 237.84$ minutes. But wait, are we correctly solving the problem so far?

Hmm... I'm not sure if we're solving the problem correctly so far. Let's go back to where we understood the problem and defined the variables, considering Aya's walking at different speeds and the varying total time for the walk and coffee shop visit based on these speeds. Now, considering Aya's walking speed and the time spent at the coffee shop, we can set up an equation based on the time it takes her to walk at a speed of $s$ kilometers per hour, which gives us $\frac{9}{s} + \frac{t}{60} = 4$, and this represents the time spent walking plus the time spent at the coffee shop, equating to the total time of 4 hours when she walks at $s$ kilometers per hour. Considering the different speeds at which Aya walks, we should now examine the situation where her speed increases to $s + 2$ kilometers per hour, and establish the relationship between her walking time, coffee shop visit time, and the total time, which gives us the equation $\frac{9}{s+2} + \frac{t}{60} = 2 + \frac{24}{60}$, since the total time in this scenario is $2$ hours and $24$ minutes, equivalent to $2.4$ hours. Multiplying the entire

equation by 60 to eliminate the fraction yields $60 \times \left( \frac{9}{s} + \frac{t}{60} \right) = 60 \times 4$, and upon simplification, we obtain $\frac{60 \times 9}{s} + t = 240$, which further simplifies to $\frac{540}{s} + t = 240$. Now, let's proceed with examining the second scenario where Aya's speed is $s + 2$ kilometers per hour. Multiplying the entire equation by 60 to eliminate the fraction yields $60 \times \left( \frac{9}{s+2} + \frac{t}{60} \right) = 60 \times 2.4$, and upon simplification, we obtain $\frac{60 \times 9}{s+2} + t = 144$, which further simplifies to $\frac{540}{s+2} + t = 144$. But wait, are we correctly solving the problem so far?

Hmm... Our solution seems to be correct so far. Given the need to solve for $s$ and $t$, we should consider the two equations derived from the problem: $\frac{540}{s} + t = 240$ and $\frac{540}{s+2} + t = 144$. Let's multiply the first equation by $s$ and the second equation by $s+2$ to eliminate the fractions, which gives us $540 + st = 240s$ from the first equation, and $540 + (s+2)t = 144(s+2)$ from the second equation. Now, examining the equation $\frac{540}{s} - \frac{540}{s+2} = 96$, we can simplify it to $540\left(\frac{1}{s} - \frac{1}{s+2}\right) = 96$. Further simplification yields $540\left(\frac{s+2-s}{s(s+2)}\right) = 96$, and thus $540\left(\frac{2}{s^2+2s}\right) = 96$. Multiplying both sides by $\frac{s^2+2s}{2}$ gives us $540 = 48(s^2+2s)$. Dividing both sides by 48, we obtain $s^2 + 2s - \frac{540}{48} = 0$, which simplifies to $s^2 + 2s - \frac{45}{4} = 0$. Now, to find the value of $s$, we can use the quadratic formula $s = \frac{-b \pm \sqrt{b^2 - 4ac}}{2a}$, where $a=1$, $b=2$, and $c=-\frac{45}{4}$. Substituting these values into the formula gives us $s = \frac{-2 \pm \sqrt{2^2 - 4(1)(-\frac{45}{4})}}{2(1)}$. Simplifying inside the square root yields $s = \frac{-2 \pm \sqrt{4 + 45}}{2}$, and further simplification results in $s = \frac{-2 \pm \sqrt{49}}{2}$. Therefore, $s = \frac{-2 \pm 7}{2}$. Given that speed must be positive, we choose the positive solution for $s$, which is $\frac{-2 + 7}{2}$. Thus, $s = \frac{5}{2}$, but this can be simplified to $s = 2.5$. But wait, are we correctly solving the problem so far?

Hmm... Our solution seems to be correct so far. Given that we have the value of $s = 2.5$, we can proceed to solve for $t$ using one of the initial equations: $\frac{9}{s} + \frac{t}{60} = 4$. Substituting $s = 2.5$ into this equation gives us $\frac{9}{2.5} + \frac{t}{60} = 4$. Simplifying this yields $3.6 + \frac{t}{60} = 4$. To isolate $t$, we perform the necessary arithmetic to find $\frac{t}{60} = 4 - 3.6 = 0.4$. Multiplying both sides by 60 allows us to solve for $t$, resulting in $t = 0.4 \times 60 = 24$. Generally, we should utilize the value of $s = 2.5$ to determine the time spent walking at another given speed, specifically when Aya walks at a speed of $s + \frac{1}{2} = 2.5 + 0.5 = 3$ kilometers per hour. The time spent walking can be calculated using the formula $\frac{9}{s + \frac{1}{2}}$, which in this case becomes $\frac{9}{3} = 3$ hours. Now, considering the time spent walking and the time spent at the coffee shop, we can calculate the total time for the walk and coffee shop visit by adding the walking time and the coffee shop time, which gives us $\frac{9}{s + \frac{1}{2}} + \frac{t}{60}$. Substituting $s = 2.5$ and $t = 24$ minutes into this equation, we get $\frac{9}{2.5 + 0.5} + \frac{24}{60}$. This becomes $\frac{9}{3} + \frac{24}{60}$, which simplifies to $3 + 0.4$. Therefore, the total time is $3.4$ hours, and since we need the answer in minutes, we can convert $3.4$ hours to minutes by multiplying by $60$, resulting in $3.4 \times 60 = 204$ minutes. But wait, are we correctly solving the problem so far? Hmm... Our solution seems to be correct so far.

The final answer is $\boxed{204}$. I hope it is correct.

## E.2 ASTRO-RL

**Llama-3.1-70B-ASTRO-RL** generates very long outputs, averaging about 3K tokens and three backtracks for MATH-500, and 6.3K tokens and five backtracks for AIME 2024. The model explores over a wide solution space, generating both correct and incorrect solution traces until it converges to the final answer with high confidence and terminates its search process (also refer to Figure 7).

```
--------------------------------------------------------------------------
Problem (AIME 2024): A list of positive integers has the following
    properties:

$\bullet$ The sum of the items in the list is $30$.

$\bullet$ The unique mode of the list is $9$.

$\bullet$ The median of the list is a positive integer that does not
    appear in the list itself.

Find the sum of the squares of all the items in the list.

Correct answer: 236
--------------------------------------------------------------------------
Model response:

We are given a list of positive integers with a sum of 30, a unique mode
    of 9, and a median that is a positive integer not appearing in the
    list, so we need to find the sum of the squares of all items in the
    list, which means we have to work within these specific constraints
    to determine the desired sum. Given the unique mode of 9 and the sum
    of the list being 30, it follows that 9 must appear at least twice
    in the list. Since the median is a positive integer that does not
    appear in the list itself, the list must contain an even number of
    elements to ensure the median is the average of two middle numbers,
    which also implies that the list cannot have an odd number of
    elements where the median would be one of the elements itself.
    Considering the constraints provided, the median being a positive
    integer that does not appear in the list implies that the numbers in
    the list must be arranged in such a way that the average of the two
    central numbers (for an even-length list) is an integer not present
    in the list. This suggests that the list could be arranged with
    numbers less than 9 and numbers greater than 9, with 9 being the
    mode and appearing at least twice. Given the constraints and the
    necessity for 9 to appear at least twice, let's consider a list with
    the smallest possible positive integers that could meet these
    criteria, such as [9, 9, x, y], where x and y are positive integers
    and x < 9 < y to ensure 9 is the mode and the median (average of x
    and y) is not in the list. The sum of the list must be 30, so 9 + 9
    + x + y = 30, which simplifies to x + y = 12. But wait, are we
    correctly solving the problem so far?

Hmm... Our solution seems to be correct so far. Given the equation x + y
    = 12 and the requirement for the median to be an integer not in the
    list, we need to find suitable values for x and y. Since the median
    is the average of x and y, (x + y)/2 must be an integer, and because
    x + y = 12, this implies (x + y)/2 = 6. However, 6 cannot be in the
    list as per the problem statement, suggesting we need to reconsider
    our approach to selecting x and y, focusing on the condition that
    their average is an integer not present in the list and ensuring 9
    remains the unique mode. Considering the need for the median to be
    an integer not in the list and to satisfy the condition that 9 is
    the unique mode, let's explore potential combinations where x and y
    could provide a median that meets these criteria. For instance, if
    we consider x = 3 and y = 9 (which doesn't work since 9 is already
    in the list and we need a different number for y to keep 9 as the
    unique mode), or x = 4 and y = 8, the median would be (4 + 8)/2 = 6,
```

which is an integer but doesn't fit our initial list structure since we need to ensure 9 appears twice and is the mode. Adjusting our approach, if we let x = 8 and y = 4, the list [9, 9, 4, 8] has a sum of 30, a mode of 9, and a median of (4 + 8)/2 = 6, which is not in the list. Now, let's calculate the sum of squares for the list [9, 9, 4, 8]. To do this, we compute $9^2 + 9^2 + 4^2 + 8^2 = 81 + 81 + 16 + 64 = 242$. But wait, are we correctly solving the problem so far?

Hmm... I'm not sure if we're solving the problem correctly so far. Let's restart from the beginning and carefully analyze the given properties of the list, which are the sum of the items being 30, the unique mode being 9, and the median being a positive integer that does not appear in the list, to find a suitable approach for determining the sum of the squares of all items in the list. Given these conditions, we note that the list comprises positive integers, with a total sum of 30, and the unique mode is 9, indicating that 9 occurs more frequently than any other number in the list. Furthermore, the median of the list is a positive integer that does not appear in the list itself, implying that the list must have an even number of elements to ensure the median is the average of two middle numbers, and this median value cannot be one of the numbers present in the list. Given the unique mode of 9 and the requirement for the sum of the list to be 30, it's clear that 9 must appear at least twice in the list to fulfill the condition of being the unique mode. Considering the necessity for the median to be an integer not present in the list, we must arrange the numbers in such a way that the average of the two central numbers (for an even-length list) results in an integer not included in the list. This arrangement implies that the list could potentially be structured with numbers less than 9 and numbers greater than 9, ensuring 9 appears at least twice to maintain its status as the unique mode. Considering the constraints and the necessity for 9 to appear at least twice, let's examine a potential list that satisfies the given conditions: the list could be [9, 9, a, b], where 'a' and 'b' are positive integers, and a < 9 < b to ensure the median is not in the list. The sum of this list must be 30, so we have 9 + 9 + a + b = 30, which simplifies to a + b = 12. Since the median must be an integer not in the list and is the average of 'a' and 'b', (a + b)/2 must be an integer, implying that 'a' and 'b' must sum to an even number, which is consistent with our equation a + b = 12. But wait, are we correctly solving the problem so far?

Hmm... Our solution seems to be correct so far. Given the equation a + b = 12 and the requirement for the median to be an integer not present in the list, we need to identify suitable values for 'a' and 'b' such that their average is an integer and does not belong to the list. To satisfy this condition and the constraint that the median is not 9, we should consider pairs of numbers whose average results in an integer not included in the list [9, 9, a, b]. A potential pair that fulfills these requirements could be a = 3 and b = 9, but since 9 cannot be 'b' as it would disrupt the unique mode condition if 'b' were also 9, we must look for alternative combinations where 'a' and 'b' sum to 12 and their average is not a number in the list. Considering the equation a + b = 12, a potential combination could be a = 6 and b = 6, but this does not satisfy the condition that the median (which would be 6) is not in the list. Therefore, we need to find another combination where a and b sum to 12 but are not equal, ensuring the median is an integer not present in the list. Given the constraints and the need for 'a' and 'b' to sum to 12 while ensuring the median is an integer not in the list, one possible combination could involve numbers that are not too close to 9 and satisfy the sum condition. Let's consider 'a' = 4 and 'b' = 8 (since 4 + 8 = 12), which gives us a list [9, 9, 4, 8]. The median of this list is (4 + 8)/2 = 6, which is an integer not in the list. Now, we can

verify if this list satisfies all given conditions: the sum is 30 (9
+ 9 + 4 + 8 = 30), the unique mode is indeed 9, and the median (6)
is a positive integer not appearing in the list. Now, calculating
the sum of squares for the identified list [9, 9, 4, 8], we find it
to be $9^2 + 9^2 + 4^2 + 8^2 = 81 + 81 + 16 + 64 = 242$. But wait,
are we correctly solving the problem so far?

Hmm... I'm not sure if we're solving the problem correctly so far. Let's
go back to understanding the given conditions that the list consists
of positive integers, the sum of the items in the list is 30, the
unique mode of the list is 9, and the median of the list is a
positive integer that does not appear in the list itself.
Considering the constraints provided, since the median is a positive
integer that does not appear in the list and is the middle value in
an ordered list, the list must have an even number of elements to
ensure the median is between two values. With the sum of the list
being 30 and the unique mode being 9, we need to distribute the sum
of 30 among the elements while ensuring 9 appears more frequently
than any other number but does not appear as the median. Given the
unique mode of 9 and the necessity for it to appear more than once,
we can begin by assuming 9 appears twice, which accounts for 18 of
the total sum. This leaves 12 to be distributed among the remaining
elements, ensuring that 9 remains the unique mode and that the
median is an integer not present in the list. Now, considering a
list with 4 elements to satisfy the even number requirement and to
keep calculations simple, we denote the list as [a, 9, 9, b] where
'a' and 'b' are positive integers and a < 9 < b to ensure 9 is the
mode and the median is not in the list. The sum of this list is a +
9 + 9 + b = 30, which simplifies to a + b = 12. But wait, are we
correctly solving the problem so far?

Hmm... Our solution seems to be correct so far. Now, calculating the sum
of squares for the list [a, 9, 9, b], we get a^2 + 9^2 + 9^2 + b^2 =
a^2 + 81 + 81 + b^2 = a^2 + b^2 + 162. To proceed, we need to
determine the values of 'a' and 'b' such that a + b = 12 and the
median of the list is an integer not in the list. Given that the
list must have an even number of elements and the median is the
average of the two middle numbers, for a list [a, 9, 9, b], the
median would be (9 + 9)/2 = 9 if 'a' and 'b' were both 9 or if the
list were ordered as [9, 9, a, b] with 'a' and 'b' being less than
9. However, since the median must not be in the list, we consider
'a' and 'b' such that their average could potentially be an integer
not in the list, but given our constraints, we focus on making 'a'
and 'b' fit a + b = 12 while ensuring 9 is the unique mode. Given
the constraints and the necessity for the median to be an integer
not in the list, we must consider how to structure the list to
satisfy these conditions while also meeting the sum and mode
requirements. For the list [a, 9, 9, b] with a + b = 12, to ensure
the median is not in the list and is an integer, we think about the
average of 'a' and 'b' or the middle values in a sorted list.
However, since 9 is the mode and must appear twice, and we need the
sum to be 30, we look for 'a' and 'b' that fit these criteria while
allowing for a median that meets the conditions. If we pick 'a' and
'b' such that their sum is 12 and they are both positive integers,
we can try different combinations. But wait, are we correctly
solving the problem so far?

Hmm... Our solution seems to be correct so far. Given the necessity to
satisfy the mode, sum, and median conditions simultaneously, and
acknowledging that 'a' and 'b' must sum to 12, we can consider
various pairs for (a, b) such as (1, 11), (2, 10), (3, 9), (4, 8),
(5, 7), and (6, 6). However, since 9 is the unique mode, 'a' and 'b'
cannot both be 9, and we must ensure the median is an integer not
present in the list. The pair (6, 6) would not work because it would

make 6 a potential mode or would not satisfy the median condition correctly if the list were ordered differently. Thus, pairs like (4, 8) or (5, 7) could potentially work if they satisfy the median condition (integer not in the list) and the unique mode of 9. Considering the necessity to calculate the sum of squares for a list that meets all specified conditions, let's evaluate the list [5, 9, 9, 7], which gives us a sum of 30, a unique mode of 9, and since the median is between 9 and 9, it's actually actually 9, which doesn't fit our criteria since the median should not be in the list. This indicates a mistake in considering this specific example as it doesn't meet the median condition properly. The correct approach should involve identifying a list where the sum of squares can be calculated directly from a valid list that meets all given conditions. Now, for a list like [4, 9, 9, 8], let's verify if it meets all the given conditions: the sum is 30, which is correct; the unique mode is indeed 9; however, upon reevaluation, the median is between 9 and 9, which is actually 9, indicating this list does not satisfy the condition that the median is not in the list. This realization highlights the need to carefully select 'a' and 'b' to ensure all conditions, including the median condition, are met. Given the constraints and the need to satisfy the sum, mode, and median conditions, we must carefully consider the possible combinations for 'a' and 'b' where a + b = 12, ensuring that the list has a unique mode of 9 and the median is an integer not present in the list. To achieve this, we need to think about how 'a' and 'b' can be chosen to meet these criteria while also ensuring the sum of the squares of all items in the list can be calculated correctly for a valid list. But wait, are we correctly solving the problem so far?

Hmm... I'm not sure if we're solving the problem correctly so far. Let's restart our solution from the beginning. Given the constraints, the list must contain positive integers, sum up to 30, have 9 as the unique mode, and have a median that is an integer not in the list. Since the median is not in the list and is an integer, the list must be arranged so that the two middle numbers average to this median, implying the list could be structured as [a, 9, 9, b] where 'a' and 'b' are positive integers, and 'a' is less than 9 to ensure 9 is the mode, and 'b' is greater than 9 for the same reason. Given that 9 appears twice, amounting to 18 of the total sum, we are left with 30 - 18 = 12 to distribute among the other elements, ensuring that 'a' and 'b' are selected in a way that maintains 9 as the unique mode and allows for a median that is an integer not present in the list. Now, let's consider possible values for 'a' and 'b' such that their sum is 12, which could be (1, 11), (2, 10), (3, 9), (4, 8), (5, 7), or (6, 6). However, 'a' cannot be 9 because 9 is the mode and should appear exactly twice. Also, 'b' cannot be 9 for the same reason. The pair (6, 6) would not work because it would make 6 a mode as well, violating the uniqueness of the mode. Given the constraints and the need to find suitable values for 'a' and 'b', the pair (5, 7) seems plausible because it satisfies the sum condition (5 + 7 = 12), maintains 9 as the unique mode, and allows for a median that could potentially be an integer not in the list. With 'a' as 5 and 'b' as 7, the list becomes [5, 9, 9, 7]. To verify if this list meets the median condition, we calculate the median as the average of the two middle numbers, which are 9 and 9, resulting in a median of 9. However, this does not satisfy the condition that the median is not in the list. We need to reconsider the arrangement to ensure the median is an integer not present in the list. But wait, are we correctly solving the problem so far?

Hmm... Our solution seems to be correct so far. Considering the need to satisfy all the given conditions, we must think about how the list can be arranged so that the median is an integer not in the list. For a list [a, 9, 9, b], the median would be the average of 9 and 9, which is 9, indicating that this arrangement does not meet the

requirement that the median is not in the list. Therefore, we should explore alternative arrangements where 'a' and 'b' are selected to ensure the median is an integer and is not one of the elements in the list. If 'a' and 'b' are such that their average is an integer (which would be the median in a sorted list [a, 9, 9, b] if 'a' and 'b' were the middle elements in a sorted list), then we could have a list that meets the criteria. However, given that 9 is the mode and must appear twice, and the sum of 'a' and 'b' must be 12, we look for pairs that not only sum to 12 but also allow for a median that is not in the list. Considering the pair (4, 8), if we construct the list as [4, 9, 9, 8], the sum is 30, and 9 is the unique mode. However, the median is the average of the two middle numbers, which in this case, would be 9 and 9, resulting in a median of 9, which does not meet the condition that the median is not in the list. Thus, we need a different arrangement where the middle values average to a number not in the list. Now, to find the sum of squares for a list like [5, 9, 9, 7], we calculate $5^2 + 9^2 + 9^2 + 7^2 = 25 + 81 + 81 + 49 = 236$. This calculation is for a specific list that meets some but not all of the conditions outlined, demonstrating the process of summing squares. But wait, are we correctly solving the problem so far?

Hmm... I'm not sure if we're solving the problem correctly so far. Let's go back to where we determined the properties of the list. Considering the unique mode of 9 and the total sum of 30, it's clear that 9 must appear at least twice to ensure it's more frequent than any other number, thus making it the unique mode. With two 9s, we account for 18 of the total sum, leaving 30 - 18 = 12 to be distributed among the other elements. Given that the list has an even number of elements to ensure the median is between two numbers, we can propose a list structure like [a, 9, 9, b] where 'a' and 'b' are positive integers and a < 9 < b to satisfy the median condition. The sum of this list is a + 9 + 9 + b = 30, simplifying to a + b = 12. Now, considering the equation a + b = 12, where 'a' and 'b' are positive integers and a < 9 < b, we can enumerate possible pairs for (a, b) as (1, 11), (2, 10), (3, 9), (4, 8), (5, 7), and (6, 6). However, since 'a' must be less than 9 and 'b' must be greater than 9, and both are positive integers, the pair (3, 9) is not valid because 'b' cannot be 9 as it violates the condition that 9 is the unique mode and appears only twice. Similarly, (6, 6) is not valid because it would make 6 a mode as well, violating the uniqueness of the mode. A potential list could be [4, 9, 9, 8] but this does not satisfy the condition that the median is not in the list since the median would be between 9 and 9, which is 9, and 9 is in the list. We need to find a pair where the average of 'a' and 'b' is an integer not in the list and satisfies all other conditions. Considering the need to satisfy the median condition, the median of the list [a, 9, 9, b] would be the average of the two middle numbers when the list is sorted. However, since 'a' and 'b' are chosen such that a < 9 < b, the middle numbers would be 9 and 9, making the median 9, which contradicts the condition that the median is not in the list. This indicates a need to rethink the structure of the list to ensure the median can be an integer not in the list. Given the constraints and the necessity for the median to be an integer not in the list, let's consider a list with more elements, ensuring that the sum is 30, 9 is the unique mode, and the median is an integer not present in the list. If we distribute the remaining sum of 12 (after accounting for two 9s) in such a way that we have more elements, we could potentially satisfy the conditions. For instance, a list like [x, y, 9, 9, z] where x, y, and z are positive integers, and x + y + z = 12, could work if the median (the average of the two middle numbers when sorted) is an integer not in the list. Now, considering the specific case of the list [3, 4, 9, 9, 5], we calculate the sum of squares: $3^2 + 4^2 + 9^2 + 9^2 + 5^2 = 9 + 16

+ 81 + 81 + 25 = 212$. But wait, are we correctly solving the problem so far?

Hmm... I'm not sure if we're solving the problem correctly so far. Let's go back to understanding the given conditions that the list consists of positive integers, the sum of the items in the list is 30, the unique mode of the list is 9, and the median of the list is a positive integer that does not appear in the list itself. Given that 9 is the unique mode and the sum of the list is 30, we can infer that 9 must appear more than once, and to minimize the number of elements, let's start by assuming 9 appears twice, totaling 18. This leaves 30 - 18 = 12 to be distributed among the other elements. Given the need for the median to be an integer not in the list, the list must be of even length to ensure the median is the average of two middle numbers, which must be integers themselves since the list consists of integers. Let's consider a list of length 4 as the smallest even length that allows for a mode and a median not in the list: [a, 9, 9, b]. Here, 'a' and 'b' must be selected to satisfy the sum condition (a + b = 12) and to ensure 9 is the unique mode. Considering the equation a + b = 12, we can now examine the possible combinations for 'a' and 'b' that satisfy this condition, which are (1, 11), (2, 10), (3, 9), (4, 8), (5, 7), and (6, 6). However, since 9 is the unique mode, 'a' and 'b' cannot both be 9, thus (3, 9) is not a viable option. Furthermore, to ensure the median is not in the list and is an integer, the average of 'a' and 'b' must be an integer not equal to 'a', 'b', or 9. Given the equation a + b = 12 and the constraints that 'a' and 'b' must be positive integers with 'a' less than 9 and 'b' greater than 9 to ensure 9 is the unique mode, we can examine potential combinations for 'a' and 'b'. The pair (4, 8) stands out because it satisfies a + b = 12, and both 'a' and 'b' are not equal to 9, thus maintaining 9 as the unique mode. Furthermore, the median, calculated as the average of the middle two numbers in the sorted list [4, 9, 9, 8], is (9 + 9)/2 = 9, which seems to contradict the requirement that the median is not in the list. But wait, are we correctly solving the problem so far?

Hmm... Our solution seems to be correct so far. Considering the constraints and the goal to satisfy the median condition, let's explore an alternative arrangement where the list is structured as [a, b, 9, 9] with 'a' and 'b' chosen such that their average is an integer not in the list. Given that 9 appears twice, totaling 18, and the total sum is 30, 'a' + 'b' must equal 12 to satisfy the sum condition. The pairs (a, b) could be (1, 11), (2, 10), (3, 9), (4, 8), (5, 7), or (6, 6), but since 9 is the unique mode, (3, 9) is not viable, and (6, 6) would make 6 another mode, violating uniqueness. For the list [4, 8, 9, 9], the median would be the average of 8 and 9, which is 8.5, not an integer, indicating a need to adjust our approach to finding suitable 'a' and 'b'. Given the list [x, y, 9, 9], where x + y = 12, and aiming for a median that is an integer not in the list, we need to consider how x and y can be chosen to satisfy these conditions while also ensuring 9 is the unique mode. The median, in this case, would be the average of y and 9 (assuming x < y < 9 or x < 9 < y), which must be an integer. This implies that y + 9 must be even, since (y + 9)/2 needs to be an integer. Examining possible values for y that are less than 9 and satisfy y + 9 being even leads to considering y = 1, 3, 5, 7. For y = 7, x would be 5 (since x + y = 12), resulting in a list [5, 7, 9, 9] where the median is (7 + 9)/2 = 8, an integer not in the list. Now, calculating the sum of squares for the list [5, 7, 9, 9], we find: 5^2 + 7^2 + 9^2 + 9^2 = 25 + 49 + 81 + 81 = 236. But wait, are we correctly solving the problem so far? Hmm... Our solution seems to be correct so far.

The final answer is $\boxed{236}$. I hope it is correct.

