# OpenReview forum: "ASTRO: Teaching Language Models to Reason by Reflecting and Backtracking In-Context"
_ICLR.cc/2026/Conference — ICLR 2026 Conference Withdrawn Submission_

### Official Review · Reviewer_uene · 2025-10-17

**Soundness:** 3
**Presentation:** 3
**Contribution:** 1
**Rating:** 2
**Confidence:** 4

**Summary:**

This paper introduces ASTRO, a method that teaches LLMs to imitate search traces derived from MCTS and then improve their policy using RL. ASTRO first constructs a search tree using MCTS and generates synthetic search traces that contain self-reflection and backtracking based on this tree. These traces are then used to fine-tune the LLM via supervised learning. Following this SFT phase, ASTRO further optimizes the LLM using GRPO. The paper shows its effectiveness on MATH-500, AMC 2023, and AIME 2024.

**Strengths:**

The paper is clearly written and easy to follow, effectively contextualizing the work within prior research such as procedure cloning [1] and stream of search [2].

[1] Yang et al., Chain of Thought Imitation with Procedure Cloning, NeurIPS 2022 \
[2] Gandhi et al., Stream of Search (SoS): Learning to Search in Language, COLM 2024

**Weaknesses:**

The paper's main objective is to investigate whether an LLM can perform long-form reasoning without relying on pre-existing reasoning traces, such as those from large CoT models like DeepSeek r1. However, I have significant concerns on this.

1. Practicality: The research objective appears misaligned with the chosen domain of mathematical reasoning. Large-scale, high-quality SFT datasets with long reasoning traces are already widely available for math tasks. Consequently, the motivation for employing a computationally expensive MCTS framework to synthesize new data remains unclear.

2. Novelty: The methodological contribution is limited. The proposed approach largely mirrors stream of search [1], replacing its BFS/DFS search algorithms with MCTS and applying it to the math domain. This modification alone offers limited algorithmic novelty.

By contrast, the NeurIPS 2025 papers Mulberry [2] and ViGoRL [3] adopted a similar MCTS-based synthesis strategy but targeted visual reasoning, a domain notably lacking long reasoning-trace datasets. These works effectively justified their approach by demonstrating a clear need for MCTS-generated data.

[1] Gandhi et al., Stream of Search (SoS): Learning to Search in Language, COLM 2024 \
[2] Yao et al., Mulberry: Empowering MLLM with o1-like Reasoning and Reflection via Collective Monte Carlo Tree Search, NeurIPS 2025 \
[3] Sarch et al., Grounded Reinforcement Learning for Visual Reasoning, NeurIPS 2025

**Questions:**

- What was the rationale for choosing a group size of 4, which seems relatively small? In addition, could you specify the PPO clipping ratio used in your experiments?
- It doesn’t seem necessary to perform backtracking only at the terminal nodes. Could you explain the reasoning behind that design choice?

---

### Official Review · Reviewer_URnr · 2025-11-01

**Soundness:** 2
**Presentation:** 2
**Contribution:** 2
**Rating:** 2
**Confidence:** 4

**Summary:**

The paper introduces ASTRO, an “Autoregressive Search-Taught Reasoner” that trains language models to internalize search-style reasoning with explicit self-reflection and backtracking. The pipeline has three stages. First, the authors run Monte Carlo Tree Search (MCTS) over stepwise math solutions to build search trees, then linearize visited paths—including detours to incorrect terminals—and rewrite them into long chain-of-thought traces that interleave forward progress with explicit “but wait…” reflections and backtracks. Second, they perform supervised fine-tuning (SFT) on ~36K such search-integrated traces to seed autoregressive search behavior. Third, they run reinforcement learning using a GRPO-style objective with verifier rewards, further encouraging longer, revision-heavy trajectories. Applied to Llama-3.1-70B, the SFT checkpoint improves over the base model on MATH-500, AMC 2023, and AIME 2024, and the RL-tuned model (ASTRO-RL) reaches 81.8% pass@1 on MATH-500 and 30.0% on AIME 2024, with increasing performance correlated to more backtracking and longer generations. The paper emphasizes that training with search-structured priors yields better RL efficacy than training on direct solutions alone.

**Strengths:**

The central idea—procedure-cloning search traces into natural language and then optimizing with verifier-based RL—is original in how it couples explicit backtracking with a purely autoregressive policy. Unlike external scaffolds, the model is trained to “think like search” within a single pass, and the linearization scheme that stitches together incorrect and correct endpoints is a clever way to distill exploration signals into text.

Methodologically, the paper is clear about the three stages, gives concrete MCTS settings, and provides prompts for both rewriting and backtracking, which enhances reproducibility. Empirically, the study separates “Direct” (no-search priors) from ASTRO, showing that search-infused SFT yields a better RL starting point and higher ceilings; the training curves and correlation analyses further support the claim that frequent reflection/backtracking is predictive of accuracy.

**Weaknesses:**

Important related work is missing or under-emphasized. In particular, Satori (an RL-trained LRM with autoregressive search behaviors), RAP, and rStar—the earliest approaches that use MCTS with LLMs—should be positioned as direct antecedents, with a careful comparison of objectives, supervision signals, tree policies, and data generation. This gap makes it harder to judge conceptual novelty relative to prior MCTS-style pipelines and RL-trained “search-in-language” agents. The evaluation suite also lags the submission year; relying on AIME 2024 in a 2026 submission risks stale estimates and potential contamination, and the work would be stronger with more contemporaneous rounds and additional leakage-controlled sets. On modeling breadth, all experiments use Llama variants; the absence of diverse base families (e.g., Qwen, Mistral) limits claims about generality of the search-prior recipe. Finally, baseline coverage is not sufficiently comprehensive for this topic: beyond Llama post-training variants, the study should include head-to-head comparisons with strong search-based reasoning agents and MCTS-trained systems under matched compute and decoding budgets. As it stands, the result table focuses on nearby Llama training methods and omits these critical baselines; expanding this comparison is necessary to establish competitiveness. These issues are addressable and would materially strengthen the paper’s case.

**Questions:**

How sensitive is ASTRO to the exact linearization policy that merges incorrect and correct terminals? For example, if you vary the number of injected backtracks k, choose different ancestor merge points, or remove the hard-coded “but wait…” scaffolding, do the SFT priors and downstream RL gains persist?

Could you ablate the self-evaluation filter as well to show whether unanimous high-quality selection is necessary for bootstrapping?

What are the compute-normalized trade-offs relative to external search scaffolds at inference time—does “internalized search” achieve similar accuracy for a fixed total token budget, and how does latency compare?

Given that performance correlates with both longer outputs and more backtracking, can you disentangle causality by constraining one while varying the other?

Can you update evaluations to newer AIME rounds and other 2025–2026 math suites to rule out temporal confounds? Clarifying these points during rebuttal would help assess robustness, generality, and the true contribution beyond prior MCTS- and RL-based reasoners.

---

### Official Review · Reviewer_xjKh · 2025-11-02

**Soundness:** 2
**Presentation:** 1
**Contribution:** 2
**Rating:** 2
**Confidence:** 3

**Summary:**

This paper proposes to use MCTS to generate reasoning trajectories, uses the generated data for warming up the model with SFT, and then train the model with RL using the verifiable rewards. The results show improvements on math reasoning tasks compared with the other RL methds.

**Strengths:**

Using MCTS to bootstrap reasoning trajectory data for SFT is a reasonable and likely to be a promising idea, if combined with RL and SFT properly. At its current version, I'm not sure if I understand the benefit of using MCTS correctly in this paper (see below).

**Weaknesses:**

- Empirical results are not strong: The main results in Table 1 only shows marginal gain over the other baselines (SPOC, StepKTO), while it's not clear how their complexity differ. My impression is that Astro is a much more complex than SPOC and Step-KTO based on the description of the paper.
- The main insight is not clear. This paper mentioned many tricks on preparing data for SFT and the application of RL, but I don't get what the main insight is compared with the prior works. As mentioned in the paper, porcedure cloning for better cold-start and then doing RL were explored before. It seems that the use of MCTS to generate SFT trajectories is the main difference to the prior works. If that's the case, it should be made clearly and deeper in the paper. In the intro (Line 81-Line 86), only one sentence mentions using MCTS to explore the solution space. I would be expecting to see the description about why MCTS is a good idea and how you deal with failed trajectories in MCTS when making SFT data.
- Related to above, when reading the paper, I feel the reason of why internalizing search behaviors to the model's generation is a good idea needs to be clarified. I was expecting to see some explaination of this around Line 44 to Line 53, but nothing was there.

**Questions:**

- Line 35: It'd be good to explain the benefit of having reasoning models iteratively refine their own outputs. It's not clear why it's better than an explicit search. Currently, it reads like "prior works said that reasoning models are good, so let's work on reasoning models".
- Line 410: The correlation between backtrack and accuracy suggested that more backtracks improves the performance, which is reasonable to some extent. However, alternatively, doesn't it also imply that the model learned to produce many incorrect answers in the first place and correct them later using backtrack? This is likely to happen because SFT data contains a lot of errored reasoning paths and the paths correcting them. It seems that Astro leads to a model generating incorrect reasoning first and correct them. This is an ideal prior to start with for RL.
- How is it compared with this work that also uses SFT+RL with linearized search?

Shen, Maohao, et al. "Satori: Reinforcement learning with chain-of-action-thought enhances llm reasoning via autoregressive search." arXiv preprint arXiv:2502.02508 (2025).

---

### Official Review · Reviewer_yqES · 2025-11-02

**Soundness:** 2
**Presentation:** 3
**Contribution:** 2
**Rating:** 4
**Confidence:** 4

**Summary:**

The paper introduces ASTRO, a three-stage framework (MCTS search trajectory generation to SFT to RL) aimed at instilling explicit self-reflection and backtracking behaviors into base LLMs for mathematical reasoning. The core novelty is incorporating artificial backtracking steps into the supervised fine-tuning (SFT) "cold-start" data, derived from Monte Carlo Tree Search (MCTS) traces.

**Strengths:**

The presentation is good and easy to understand. The figures elucidate the concepts, particularly the structure of the MCTS traces and the linearization process.

The main novelty is introducing the artificial backtracking behaviors. This differentiates from many prior works which also collect MCTS traces but only use directed correct paths.

The experiments demonstrate that models initialized with the search prior outperform those initialized only with direct solutions. The training and evaluation mostly follow standard practices.

**Weaknesses:**

We know that reasoning models extensively use backtracking, a key behavior that previous instruction following models don't have. It is good to know that artificial backtracking improves performance, but from this point of view, the novelty appears limited. It would be better to ablate on this key novelty. Even though we publicly know very little about deepseek-r1's cold-start data, or whether openai-o1 uses cold-start data at all, the author could compare the effect of their synthetic data with data generated from from a reasoning model and used as cold-start data for training instruction following models.

The performance gain in table 2 against a model trained on direct data from the same MCTS trees is rather limited in my opinion, and the authors could better situate this gain by detailing what is the computational cost of the procedural cloning step in section 2.4. MCTS is also known to be very costly, therefore a comparison with direct data from the simpler best-of-N sampling is also a nice-to-have baseline.

The authors only generate and train with 1 model, namely llama3.1-70B. Does the same pipeline work with smaller (7B) models? Are the self-evaluation (line 188) and few-shot prompt (line 220) all done by llama3.1-70B?

**Questions:**

Please see the weaknesses section.

---

### Note · Authors · 2025-11-21

**Comment:**

We thank the reviewers for their constructive feedback.

**Withdrawal Confirmation:**

I have read and agree with the venue's withdrawal policy on behalf of myself and my co-authors.